# An umbrella review of the evidence linking oral health and systemic noncommunicable diseases

João Botelho ⬤[1,2] ✉, Paulo Mascarenhas ⬤[2], João Viana[1], Luís Proença ⬤[1,2], Marco Orlandi[3], Yago Leira[3,4,5], Leandro Chambrone ⬤[2,6,7], José João Mendes[1,2] & Vanessa Machado ⬤[1,2]

Oral diseases are highly prevalent worldwide. Recent studies have been supporting a potential bidirectional association of oral diseases with systemic noncommunicable diseases (NCDs). Available evidence supports that people with NCDs have a greater prevalence of oral diseases particularly those with limited ability of oral self-care. Regarding the reverse relationship, the lines of evidence pointing out NCDs as putative risk factors for oral diseases have increased significantly but not with a consistent agreement. This umbrella review of meta-analyses appraises the strength and validity of the evidence for the association between oral health and systemic health (registered at PROSPERO, ID: CRD42022300740). An extensive search included systematic reviews that have provided meta-analytic estimates on the association of oral diseases with NCDs. The overall strength of evidence was found to be unfavorable and with methodological inconsistencies. Twenty-eight NCDs were strongly associated with oral diseases. Among those NCDs are five types of cancer, diabetes mellitus, cardiovascular diseases, depression, neurodegenerative conditions, rheumatic diseases, inflammatory bowel disease, gastric helicobacter pylori, obesity, and asthma. According to fail-safe number statistics, the evidence levels are unlikely to change in the future, indicating a fairly robust consistency.

Oral diseases are chronic and progressive conditions that affect the health of teeth and mouth[1,2]. Beyond its pronounced worldwide prevalence and a clear public health concern, oral diseases have been proposed to have a bidirectional association with systemic health only suggested in recent years[3–11]. While evidence of this bidirectional link is robust in diseases that limit oral self-care (either physical or cognitive incapacity), the association of oral diseases with other chronic noncommunicable diseases (NCDs) has increased, still without the proper consistency.

The World Health Organization (WHO) approved, in 2021, a Resolution on oral health, urging key risk factors of oral diseases shared with other NCDs[12]. Instead of the traditional curative approach, WHO caveats the importance of prevention encompassing oral health in universal health coverage programs. Over 3.5 billion people are

[1]Clinical Research Unit, Centro de Investigação Interdisciplinar Egas Moniz (CiiEM), Egas Moniz—Cooperativa de Ensino Superior, CRL, Almada, Portugal. [2]Evidence-Based Hub, Centro de Investigação Interdisciplinar Egas Moniz (CiiEM), Egas Moniz—Cooperativa de Ensino Superior, CRL, Almada, Portugal. [3]Periodontology Unit, UCL Eastman Dental Institute and NIHR UCLH Biomedical Research Centre, University College London, London, UK. [4]Periodontology Unit, Faculty of Medicine and Odontology, University of Santiago de Compostela, Santiago, Spain. [5]Clinical Neurosciences Research Laboratory, Health Research Institute of Santiago de Compostela (IDIS), Santiago de Compostela, Spain. [6]Unit of Basic Oral Investigation (UIBO), Universidad El Bosque, Bogota, Colombia. [7]Department of Periodontics, University of Iowa College of Dentistry, Iowa City, IA, USA. ✉e-mail: jbotelho@egasmoniz.edu.pt

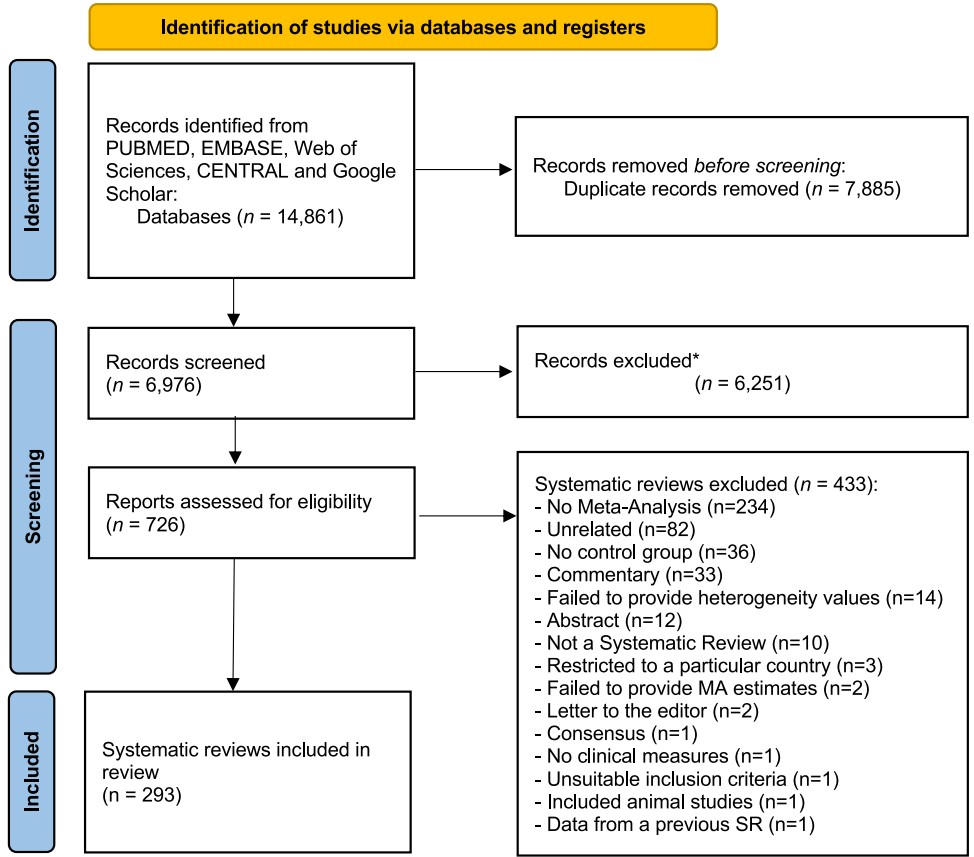

**Identification of studies via databases and registers**

**Identification**

Records identified from PUBMED, EMBASE, Web of Sciences, CENTRAL and Google Scholar:
Databases (*n* = 14,861)

→ Records removed *before screening*:
Duplicate records removed (*n* = 7,885)

**Screening**

Records screened (*n* = 6,976)

→ Records excluded* (*n* = 6,251)

Reports assessed for eligibility (*n* = 726)

→ Systematic reviews excluded (*n* = 433):
- No Meta-Analysis (n=234)
- Unrelated (n=82)
- No control group (n=36)
- Commentary (n=33)
- Failed to provide heterogeneity values (n=14)
- Abstract (n=12)
- Not a Systematic Review (n=10)
- Restricted to a particular country (n=3)
- Failed to provide MA estimates (n=2)
- Letter to the editor (n=2)
- Consensus (n=1)
- No clinical measures (n=1)
- Unsuitable inclusion criteria (n=1)
- Included animal studies (n=1)
- Data from a previous SR (n=1)

**Included**

Systematic reviews included in review (n = 293)

**Fig. 1 | PRISMA Flowchart.** Flow diagram visually summarising the screening and selection processes, and the numbers of articles recorded at each different stage.

estimated to suffer from oral diseases, and the associated burden is likely to remain or increase, particularly during the current global pandemic, with successive lockdowns that have limited access to oral health care. Conducting an extensive analysis on the degree of evidence of the association of oral diseases with NCDs becomes highly relevant to inform and influence public health and health policy-makers. For this reason, we aimed to perform an umbrella review to overlook the robustness of the meta-analytic estimates linking oral and systemic diseases and its bidirectional association. We additionally aimed to explore whether future research will likely transform the inferences from existing significant meta-analyses.

Herein, our results show that 28 NCDs, including five types of cancer, and circulating levels of CRP were strongly associated with oral diseases. Most evidence is unlikely to change in the future, with a few exceptions, due to fairly robust evidence consistency.

## Results

### Selection and characteristics of the included meta-analyses

Our search retrieved a total of 14,861 entries (Fig. 1). After removing duplicates (*n* = 7885), a total of 6976 records were screened for title and abstracts against the eligibility criteria. After judging the full paper of 726 records, 433 studies were excluded (the list of excluded studies with justification for exclusion is detailed in Supplementary Data 1). Excellent inter-examiner reliability was confirmed at the full-text screening (Cohen's kappa score = 0.91, 95% CI: 0.88; 0.93). A final sample of 293 systematic reviews with meta-analyses were included for further appraisal (Supplementary Data 2).

Most systematic reviews were conducted in China (*n* = 93), Brazil (*n* = 47), USA (*n* = 22), UK (*n* = 21), Spain (*n* = 15) and Italy (*n* = 13); still we observed studies from a substantial number of countries (Supplementary Data 3). The majority followed Preferred Reporting Items for Systematic Reviews and Meta-Analysis (PRISMA) (*n* = 186, 63.5%),

Meta-analyses Of Observational Studies in Epidemiology (MOOSE) (*n* = 21, 7.2%), or using more than one guideline (*n* = 6, 2.0%). Yet, fourteen did not report following a reporting guideline for systematic reviews (*n* = 23, 7.9%). While for risk of bias (methodological quality), Newcastle-Ottawa Scale (NOS) (*n* = 133, 45.4%), Cochrane tools (*n* = 50, 17.1%) or Joanna Briggs Institute (JBI) tools (*n* = 18, 6.1%) were the most used instruments.

Overall, 855 meta-analytic comparisons were included. More than half meta-analyses (*n* = 448; 52.4%) used a continuous exposure contrast, whereas the remaining used a binary analysis (Supplementary Data 2). Mean Difference (31.9%, *n* = 273), Odds Ratio (28.3%, *n* = 242), Risk Ratio (16.0%, *n* = 137) and Standardized Mean Difference (14.5%, *n* = 124) were the most common reported effect measures (Supplementary Data 2). Out of the 293 studies, 69.2% (*n* = 203) were published between 2011 and 2020, while 26.3% (*n* = 77) were published in 2021 and 2022, and 4.8% (*n* = 14) until 2010. About 24.5% (*n* = 72) had a search period limit of 2020 to 2022. Most meta-analyses used oral diseases and/or treatments as an exposure (*n* = 485, 56.7%), while the remaining used them as an outcome. The summary descriptive characteristics of the included meta-analyses by oral condition is presented in Table 1.

### Summary effects and heterogeneity between studies

Of the 855 meta-analytic comparisons, 592 (69.2%) were nominally significant ($p < 0.05$), with only 7.7% of strong meta-analytical evidence (*n* = 66), while 18.4% (*n* = 157) and 6.4% (*n* = 55) were of highly suggestive and suggestive evidence, respectively (Table 1). Of the stricter *P*-value threshold, 355 (41.5%) and 120 (14.0%) meta-analyses had significance at $10^{-6}$ and $10^{-3}$, respectively. Approximately 62.5% (*n* = 534) of the included meta-analyses had high heterogeneity ($I^2 > 50\%$), with 27.7% (*n* = 237) of them presenting low heterogeneity ($I^2 \leq 25\%$).

**Table 1 | Descriptive statistics and evidence grading of the included meta-analyses of oral conditions**

| | Dental Caries | Periodontal disease | Tooth loss | Edentulism | Endodontic conditions | Dental trauma | Tooth erosion | Dental implant conditions | Oral lesions | Bruxism | Others[a] |
|---|---|---|---|---|---|---|---|---|---|---|---|
| Number of meta-analyses | 65 | 487 | 33 | 15 | 17 | 6 | 3 | 14 | 16 | 4 | 17 |
| Number of studies, median (min-max) | 5 (2–32) | 6 (2–65) | 7 (2–19) | 4.5 (2–33) | 3 (2–10) | 4 (2–7) | 9 (9–10) | 7 (2–14) | 2 (2–11) | 17 (2–23) | 6 (2–32) |
| Number of participants, median (min-max) | 812 (124–22,338) | 1001 (42–74,113,394) | 20,534 (246–5,507,766) | 31,855 (1,305–5,592,964) | 310 (80–670,714) | 1234 (380–5659) | 1112 (300–1550) | 676 (226–12,736) | 491 (402–12,391) | 3819 (180–4673) | 675 (96–10,064) |
| **Meta-analytical criterion, n (%)** | | | | | | | | | | | |
| $P$ value $<10^{-6}$, $n$ (%) | 18 (27.7) | 235 (48.3) | 21 (63.6) | 12 (80.0) | 8 (47.1) | 1 (16.7) | 2 (66.7) | 7 (50.0) | 8 (50.0) | 3 (75.0) | 7 (41.2) |
| $P$ value $<10^{-3}$, $n$ (%) | 8 (12.3) | 73 (15.0) | 6 (18.2) | 1 (6.7) | 3 (17.6) | 1 (16.7) | 1 (33.3) | 0 (0) | 0 (0) | 1 (25.0) | 3 (17.7) |
| $P$ value $<0.05$, $n$ (%) | 9 (13.9) | 61 (12.5) | 1 (3.0) | 2 (13.3) | 1 (5.9) | 3 (50.0) | 0 (0) | 7 (50.0) | 0 (0) | 0 (0) | 4 (23.52) |
| $I^2 >50\%$, $n$ (%) | 46 (70.8) | 307 (63.0) | 24 (72.7) | 12 (80.0) | 7 (41.2) | 3 (50.0) | 2 (66.7) | 7 (50.0) | 10 (62.5) | 3 (75.0) | 12 (70.6) |
| $I^2 \leq25\%$, $n$ (%) | 13 (20.0) | 133 (27.3) | 3 (9.1) | 3 (20.0) | 8 (47.1) | 3 (50.0) | 0 (0) | 3 (21.4) | 6 (37.5) | 1 (25.0) | 5 (29.4) |
| **Overall grading, n (%)** | | | | | | | | | | | |
| Not significant | 30 (46.2) | 118 (24.2) | 5 (15.2) | 0 (0) | 5 (29.4) | 1 (16.7) | 0 (0) | 5 (35.7) | 8 (50.0) | 0 (0) | 3 (17.6) |
| Weak | 21 (32.3) | 171 (35.1) | 2 (6.1) | 3 (20.0) | 10 (58.8) | 4 (66.7) | 1 (33.3) | 9 (64.3) | 7 (43.8) | 1 (25.0) | 9 (52.9) |
| Suggestive | 3 (4.6) | 44 (9.0) | 5 (15.2) | 0 (0) | 0 (0) | 0 (0) | 0 (0) | 0 (0) | 0 (0) | 0 (0) | 1 (5.9) |
| Highly suggestive | 7 (10.8) | 107 (22.0) | 13 (39.4) | 10 (66.7) | 1 (5.9) | 0 (0) | 2 (66.7) | 0 (0) | 1 (6.3) | 3 (75.0) | 2 (11.8) |
| Strong | 4 (6.2) | 47 (9.7) | 8 (24.2) | 2 (13.3) | 1 (5.9) | 1 (16.7) | 0 (0) | 0 (0) | 0 (0) | 0 (0) | 2 (11.8) |

[a]This group includes results from denture stomatitis, mouth breathing, salivary conditions and salivary flow rate.

## Grading of the evidence from oral diseases

Sixty-six meta-analyses (7.7%) were categorized as of strong evidence (Figs. 2–5), 35 with oral diseases as exposure towards a particular NCD and 19 with oral diseases as outcome. Considering oral diseases as an exposure of a particular NCD, the following associations were found: dental caries with iron deficiency ($n = 2$) (Fig. 2); tooth loss with cognitive impairment ($n = 2$), with dementia ($n = 3$) and with lung cancer ($n = 1$) (Fig. 2); edentulism with pancreatic cancer ($n = 1$) (Fig. 2); endodontic infection with serum C-reactive protein (CRP) (Fig. 3). In addition, periodontal disease presented strong association with higher risk (Fig. 4) towards: cancer ($n = 9$); cardiovascular disease (CVD) ($n = 5$); diabetes mellitus ($n = 3$); adverse pregnancy outcomes (APOs) ($n = 2$); lower longevity ($n = 2$); neurodegeneration (cognitive impairment and dementia) ($n = 3$); polycystic ovarian syndrome (PCOS) ($n = 1$); psoriasis ($n = 1$); altered levels of mean corpuscular hemoglobin (MCH) ($n = 1$); and, with serum CRP ($n = 1$). Comparing with healthy counterparts, the following associations were found considering oral condition as outcome: mental disorders with dental caries ($n = 2$) and tooth loss ($n = 1$) (Fig. 2); conditions of special needs with dental trauma ($n = 1$) (Fig. 2); diabetes mellitus with denture stomatitis ($n = 1$) (Supplementary Data 2) and periodontal disease ($n = 2$) (Fig. 5); CVD with higher average tooth loss ($n = 1$) (Fig. 2) and periodontitis ($n = 2$) (Fig. 5); asthma with mouth breathing ($n = 1$) (Supplementary Data 2) and higher average of gingival bleeding ($n = 1$) (Fig. 5); and, patients with obesity with edentulism ($n = 1$) (Fig. 2). As well, this risk towards periodontal disease (Fig. 5) was found associated with: lower physical activity ($n = 1$) (shown in figure that high physical activity is associated to less odds of having periodontal disease); rheumatoid arthritis ($n = 1$); PCOS ($n = 1$); nonalcoholic fatty liver disease ($n = 1$); medication-related osteonecrosis of the jaw (MRONJ) ($n = 1$); ankylosing spondylitis ($n = 1$); inflammatory bowel disease ($n = 2$); obstructive sleep apnea ($n = 1$) and, gastric *helicobacter pylori* ($n = 3$).

## Grading of the evidence from the impact of oral treatments

A total of 179 meta-analyses (20.9%) explored the impact of an oral treatment (either periodontal, endodontic, dental treatments in general or mandibular advancement) on NCDs and/or makers (Table 2). Only two had strong meta-analytical evidence: periodontal treatment on systemic inflammation ($n = 1$); and endodontic treatment on CVDs ($n = 1$) (Fig. 6 displays the map of evidence on these associations). Furthermore, periodontal treatment presented highly suggestive evidence of impacting CVDs ($n = 1$), metabolic disorders ($n = 3$), APOs ($n = 3$), respiratory diseases ($n = 1$) and systemic inflammation ($n = 3$), as well as suggestive evidence on APOs ($n = 3$). In addition, weak evidence was found on the effect of periodontal therapy on APOs ($n = 5$), blood levels ($n = 1$), CVDs ($n = 3$), chronic kidney disease (CKD) ($n = 1$), gastrointestinal traits ($n = 6$), metabolic disorders ($n = 32$), rheumatic diseases ($n = 10$) and systemic inflammation ($n = 10$).

## Methodological quality assessment

Good inter-examiner reliability at the A Measurement Tool to Assess Systematic Reviews 2 (AMSTAR 2) screening was recorded (Cohen kappa score = 0.84; 95% confidence interval (CI): 0.81–0.88). Only nineteen meta-analyses were conducted with high methodological quality (6.5%) and eleven with moderate (3.7%), according to this appraisal tool (Supplementary Data 4). The majority presented low ($n = 54$, 18.4%) to critically low methodological quality ($n = 207$, 71.4%). The included meta-analyses predominantly failed to report on the funding sources for the studies included in the review ($n = 272$, 93.5%), to assess the potential impact of risk of bias in the meta-analysis ($n = 215$, 73.9%), to list the excluded studies with the respective justification ($n = 207$, 71.1%) and to account the risk of bias in the interpretation and discussion of the results ($n = 175$, 60.1%). At a lower proportion, but also seriously, a comprehensive literature search strategy was lacking in 37.8% of the included studies ($n = 110$), while the

| | Decreased risk | | | | Non-significant | Increased risk | | | |
|---|---|---|---|---|---|---|---|---|---|
| | Strong | Highly suggestive | Suggestive | Weak | | Weak | Suggestive | Highly suggestive | Strong |
| **Dental caries as exposure** | | | | | | | | | |
| Adverse Pregnancy Outcomes | 0 | 0 | 0 | 0 | 1 | 0 | 0 | 0 | 0 |
| Blood Disorders | 0 | 0 | 0 | 0 | 4 | 4 | 0 | 0 | 2 |
| **Dental caries as outcome** | | | | | | | | | |
| Adverse Pregnancy Outcomes | 0 | 0 | 0 | 0 | 2 | 0 | 0 | 0 | |
| Cardiovascular Diseases | 0 | 0 | 0 | 0 | 0 | 1 | 1 | 0 | |
| Chronic Kidney Disease | 0 | 0 | 0 | 0 | 2 | 0 | 0 | 0 | |
| Gastrointestinal tract | 0 | 0 | 0 | 0 | 0 | 2 | 0 | 0 | |
| Marfan's Syndrome | 0 | 0 | 0 | 0 | 1 | 0 | 0 | 0 | |
| Mental Disorders | 0 | 0 | 0 | 0 | 9 | 6 | 2 | 4 | 2 |
| Metabolic Disorders | 0 | 0 | 0 | 0 | 7 | 4 | 0 | 2 | 0 |
| Neurodegenerative Diseases | 0 | 0 | 0 | 0 | 0 | 0 | 0 | 1 | 0 |
| Oral Cleafts | 0 | 0 | 0 | 0 | 2 | 3 | 0 | 0 | 0 |
| Rheumatic diseases | 0 | 0 | 0 | 0 | 2 | 1 | 0 | 0 | 0 |
| **Edentulism as exposure** | | | | | | | | | |
| Adverse Pregnancy Outcomes | 0 | 0 | 0 | 0 | 0 | 1 | 0 | 1 | 2 |
| Blood Disorders | 0 | 0 | 0 | 0 | 0 | 0 | 0 | 1 | 0 |
| Gastrointestinal tract | 0 | 0 | 0 | 0 | 0 | 0 | 0 | 1 | 0 |
| Marfan's Syndrome | 0 | 0 | 0 | 0 | 0 | 1 | 0 | 5 | 0 |
| **Edentulism as outcome** | | | | | | | | | |
| Metabolic Disorders | 0 | 0 | 0 | 0 | 0 | 1 | 0 | 0 | 0 |
| Neurodegenerative Diseases | 0 | 0 | 0 | 0 | 0 | 0 | 0 | 1 | 0 |
| Oral Cleafts | 0 | 0 | 0 | 0 | 0 | 0 | 0 | 0 | 1 |
| Rheumatic diseases | 0 | 0 | 0 | 0 | 0 | 0 | 0 | 1 | 0 |
| **Tooth Loss as exposure** | | | | | | | | | |
| Cancer | 0 | 0 | 0 | 0 | 0 | 0 | 1 | 1 | 1 |
| Cardiovascular Diseases | 0 | 0 | 0 | 0 | 1 | 0 | 0 | 1 | 0 |
| Metabolic disorders | 0 | 0 | 0 | 0 | 0 | 0 | 1 | 0 | 0 |
| Mortality | 0 | 0 | 0 | 0 | 1 | 0 | 0 | 3 | 0 |
| Neurodegenerative Diseases | 0 | 0 | 0 | 0 | 0 | 0 | 0 | 2 | 5 |
| **Tooth Loss as outcome** | | | | | | | | | |
| Cardiovascular Diseases | 0 | 0 | 0 | 0 | 1 | 0 | 1 | 0 | 1 |
| Mental Disorders | 0 | 0 | 0 | 0 | 0 | 1 | 1 | 1 | 1 |
| Metabolic Disorders | 0 | 0 | 0 | 0 | 0 | 0 | 1 | 4 | 0 |
| Neurodegenerative Diseases | 0 | 0 | 0 | 0 | 2 | 0 | 0 | 0 | 0 |
| Oral Cleafts | 0 | 0 | 0 | 0 | 0 | 1 | 0 | 0 | 0 |
| Respiratory diseases | 0 | 0 | 0 | 0 | 0 | 0 | 0 | 1 | 0 |

**Fig. 2 | Evidence grading diagram on dental caries, edentulism and tooth loss (both as exposure and outcome) of an NCD.** The right side displays associations that increase the risk for the respective NCD (in red), whereas the left side shows associations that reduce the risk (in green). APO adverse pregnancy outcomes, CVDs cardiovascular diseases, CKD chronic kidney disease, Dis. disease. Source data are provided as a Source Data file.

definition of the review methods a priori was not accounted for in 18.2% (n = 53). The studies selection and data extraction in duplicates was not observed in 18.6% (n = 54) and 29.6% (n = 86), respectively. Publication bias was not performed in 22.7% (n = 66) of meta-analyses, even if required.

## Number of additional studies needed to change current meta-analytic evidence

Among the 294 meta-analyses that achieved suggestive to strong evidence, the median fail-safe number (FSN) was 51 with a notable wide range of values (range: 1–87,973). For each level of evidence, the median FSN was 15 (range: 1–83) for suggestive, 65 (range: 2–87,973) for highly suggestive and 60 (range: 5–1618) for strong evidence (Supplementary Data 5). The FSN was higher than the number of studies included in 97.5% of the meta-analyses (n = 273) for these evidence categories, meaning that the statistical significance of the summary estimates is highly unlikely to change as studies are further added in the future. Regarding the 307 weak evidence meta-analyses, the median FSN was 12 (range: 0–1501), with the FSN being smaller than the number of included studies in the existing meta-analyses in 86 comparisons (28.0%).

## Discussion

The present umbrella review assessed a total of 294 meta-analyses with a total sample of 856 comparisons. Fifty-nine associations were considered of strong evidence, supported by highly significant results. The available evidence allowed us to group strong evidence into three main categories: (i) people affected by NCDs at a higher risk towards oral diseases; (ii) the exposure of an oral disease increasing the risk towards a NCD; (iii) the systemic effect of oral interventions. Fourteen NCDs were associated with a higher risk of having an oral disease: depression (with dental caries and tooth loss), several mental disorders (with dental caries), rheumatoid arthritis (with periodontal disease), ankylosing spondylitis (with periodontal disease), inflammatory bowel disease (with periodontal disease), nonalcoholic fatty liver disease (NAFLD) (with periodontal disease), PCOS (with periodontal disease), MRONJ (with periodontal disease), gastric helicobacter pylori (with periodontal disease), obstructive sleep apnea (with periodontitis), stroke (with tooth loss), obesity (with edentulism), diabetes mellitus (with denture stomatitis), asthma (with periodontal disease and mouth breathing) and various conditions of special needs (with dental trauma). Physically active people were associated with a lower likelihood towards periodontitis. Regarding the role of oral diseases on systemic NCDs, most associations pertained to periodontal diseases with APOs, diabetes mellitus, gestational diabetes mellitus, CKD, CVD, PCOS, dementia, psoriasis, cancer (breast, pancreas, prostate, lung, head and neck), cognitive decline and dementia. Other associations, particularly tooth loss and edentulism (with pancreatic and lung cancer, cognitive decline and dementia), and dental caries with iron deficiency. In

| | Decreased risk | | | | Non-significant | Increased risk | | | |
|---|---|---|---|---|---|---|---|---|---|
| | Strong | Highly suggestive | Suggestive | Weak | | Weak | Suggestive | Highly suggestive | Strong |
| **Endodontic infection as exposure** | | | | | | | | | |
| Cardiovascular Diseases | 0 | 0 | 0 | 0 | 1 | 0 | 0 | 0 | 0 |
| Systemic Inflammation | 0 | 0 | 0 | 0 | 3 | 6 | 0 | 0 | 1 |
| **Endodontic infection as outcome** | | | | | | | | | |
| Cardiovascular Diseases | 0 | 0 | 0 | 0 | 0 | 0 | 0 | 1 | 0 |
| Metabolic Disorders | 0 | 0 | 0 | 0 | 0 | 4 | 0 | 0 | 0 |
| Viral Infection | 0 | 0 | 0 | 0 | 1 | 0 | 0 | 0 | 0 |
| **Dental Implant conditions as exposure** | | | | | | | | | |
| Metabolic Disorders | 0 | 0 | 0 | 0 | 2 | 0 | 0 | 0 | 0 |
| **Dental Implant conditions as outcome** | | | | | | | | | |
| Metabolic Disorders | 0 | 0 | 0 | 0 | 3 | 9 | 0 | 0 | 0 |
| **Oral Lesions as exposure** | | | | | | | | | |
| Metabolic disorders | 0 | 0 | 0 | 0 | 1 | 0 | 0 | 0 | 0 |
| Transplanted recipients | 0 | 0 | 0 | 0 | 0 | 0 | 0 | 1 | 0 |
| **Oral Lesions as outcome** | | | | | | | | | |
| Chronic Kidney Disease | 0 | 0 | 0 | 0 | 1 | 0 | 0 | 0 | 0 |
| Gastrointestinal tract | 0 | 0 | 0 | 0 | 3 | 0 | 0 | 0 | 0 |
| Metabolic Disorders | 0 | 0 | 0 | 0 | 2 | 0 | 0 | 0 | 0 |
| Viral Infection | 0 | 0 | 0 | 0 | 1 | 0 | 0 | 0 | 0 |
| **Dental trauma as outcome** | | | | | | | | | |
| Epilepsy | 0 | 0 | 0 | 0 | 0 | 1 | 0 | 0 | 0 |
| Mental Disorders | 0 | 0 | 0 | 0 | 1 | 3 | 0 | 0 | 0 |
| Special Needs | 0 | 0 | 0 | 0 | 0 | 0 | 0 | 0 | 1 |

**Fig. 3 | Evidence grading diagram on endodontic infection, dental implant conditions, oral lesions and dental (both as exposure and outcome) of an NCD.** The right side displays associations that increase the risk for the respective NCD (in red), whereas the left side shows associations that reduce the risk (in green). APO adverse pregnancy outcomes, CVDs cardiovascular diseases, CKD chronic kidney disease, Dis. disease. Source data are provided as a Source Data file.

| | Decreased risk | | | | Non-significant | Increased risk | | | |
|---|---|---|---|---|---|---|---|---|---|
| | Strong | Highly suggestive | Suggestive | Weak | | Weak | Suggestive | Highly suggestive | Strong |
| **Periodontal diseases as outcome** | | | | | | | | | |
| Adverse Pregnancy Outcomes | 0 | 0 | 0 | 0 | 1 | 5 | 0 | 0 | 0 |
| Blood Disorders | 0 | 0 | 0 | 0 | 0 | 3 | 1 | 1 | 0 |
| Cancer | 0 | 0 | 0 | 0 | 0 | 2 | 0 | 0 | 0 |
| Cardiovascular Diseases | 0 | 0 | 0 | 0 | 3 | 6 | 3 | 7 | 2 |
| Chronic Kidney Disease | 0 | 0 | 0 | 0 | 2 | 8 | 2 | 1 | 0 |
| Emotional Disorders | 0 | 0 | 0 | 0 | 0 | 0 | 0 | 2 | 0 |
| Gastrointestinal tract | 0 | 0 | 0 | 0 | 0 | 6 | 0 | 5 | 5 |
| Liver Disease | 0 | 0 | 0 | 0 | 2 | 0 | 1 | 0 | 1 |
| Macular degeneration | 0 | 0 | 0 | 0 | 0 | 1 | 0 | 0 | 0 |
| Marfan's Syndrome | 0 | 0 | 0 | 0 | 3 | 0 | 0 | 0 | 0 |
| Mental Disorders | 0 | 0 | 0 | 0 | 6 | 3 | 0 | 1 | 0 |
| Metabolic Disorders | 0 | 0 | 0 | 0 | 9 | 14 | 3 | 13 | 2 |
| MRONJ | 0 | 0 | 0 | 0 | 0 | 0 | 0 | 0 | 1 |
| Neurodegenerative Diseases | 0 | 0 | 0 | 0 | 0 | 14 | 2 | 0 | 0 |
| Oral clefts | 0 | 0 | 0 | 0 | 0 | 2 | 0 | 1 | 0 |
| Osteoporosis | 0 | 0 | 0 | 0 | 0 | 0 | 0 | 1 | 0 |
| Physical activity | 0 | 0 | 0 | 0 | 0 | 0 | 0 | 0 | 0 |
| Psoriasis | 0 | 0 | 0 | 0 | 2 | 4 | 0 | 1 | 0 |
| Polycystic Ovary Syndrome | 1 | 0 | 1 | 0 | 0 | 2 | 0 | 0 | 0 |
| Respiratory diseases | 0 | 0 | 0 | 0 | 5 | 13 | 0 | 1 | 2 |
| Rheumatic diseases | 0 | 0 | 0 | 0 | 14 | 5 | 2 | 0 | 2 |
| Sexual Health Conditions | 0 | 0 | 0 | 0 | 0 | 0 | 0 | 3 | 1 |
| Systemic Inflammation | 0 | 0 | 0 | 0 | 0 | 2 | 0 | 0 | 0 |
| Transplanted recipients | 0 | 0 | 0 | 0 | 1 | 5 | 0 | 0 | 0 |
| Viral Infection | 0 | 0 | 0 | 0 | 4 | 11 | 1 | 3 | 0 |

**Fig. 4 | Evidence grading diagram on periodontal diseases as an outcome of a respective NCD.** The right side displays associations that increase the risk for the respective systemic NCD (in red), whereas the left side shows associations that reduce the risk (in green). APOs adverse pregnancy outcomes, CVDs cardiovascular diseases, CKD chronic kidney disease, Dis. disease. Source data are provided as a Source Data file.

addition, both periodontal and endodontic infections were confirmed as sources of systemic inflammation, as well periodontal disease was linked to changes in MCH. Regarding intervention evidence, periodontal and endodontic procedures were strongly associated with improvements in circulating levels of CRP and lower risk towards CVD, respectively. All in all, a total of 28 NCDs, including 5 types of cancer, and circulating markers of inflammation were strongly associated with oral diseases.

| | Decreased risk | | | | Non-significant | Increased risk | | | |
|---|---|---|---|---|---|---|---|---|---|
| | Strong | Highly suggestive | Suggestive | Weak | | Weak | Suggestive | Highly suggestive | Strong |
| **Periodontal diseases as exposure** | | | | | | | | | |
| Adverse Pregnancy Outcomes | 0 | 0 | 0 | 0 | 9 | 3 | 2 | 13 | 2 |
| Blood Disorders | 0 | 0 | 0 | 0 | 28 | 28 | 2 | 3 | 1 |
| Cancer | 0 | 0 | 0 | 0 | 15 | 4 | 12 | 7 | 9 |
| Cardiovascular Diseases | 0 | 0 | 0 | 0 | 3 | 7 | 4 | 10 | 4 |
| Chronic Kidney Disease | 0 | 0 | 0 | 0 | 0 | 2 | 0 | 1 | 2 |
| Mental Disorders | 0 | 0 | 0 | 0 | 0 | 0 | 0 | 2 | 0 |
| Metabolic Disorders | 0 | 0 | 0 | 0 | 6 | 6 | 5 | 11 | 3 |
| Mortality | 0 | 0 | 0 | 0 | 0 | 0 | 0 | 2 | 2 |
| Neurodegenerative Diseases | 0 | 0 | 0 | 0 | 0 | 4 | 0 | 4 | 3 |
| Psoriasis | 0 | 0 | 0 | 0 | 0 | 0 | 0 | 0 | 1 |
| Respiratory diseases | 0 | 0 | 0 | 0 | 1 | 4 | 0 | 1 | 0 |
| Rheumatic diseases | 0 | 0 | 0 | 0 | 1 | 3 | 1 | 3 | 0 |
| Sexual Health Conditions | 0 | 0 | 0 | 0 | 0 | 0 | 0 | 3 | 1 |
| Stress | 0 | 0 | 0 | 0 | 1 | 0 | 0 | 0 | 0 |
| Systemic Inflammation | 0 | 0 | 0 | 0 | 1 | 3 | 0 | 6 | 1 |
| Transplanted recipients | 0 | 0 | 0 | 0 | 0 | 1 | 0 | 0 | 0 |
| Viral Infection | 0 | 0 | 0 | 0 | 1 | 0 | 2 | 0 | 0 |

**Fig. 5 | Evidence grading diagram on periodontal diseases as an exposure to an NCD.** The right side displays associations that increase the risk for the respective NCD (in red), whereas the left side shows associations that reduce the risk (in green). APOs adverse pregnancy outcomes, CVDs cardiovascular diseases, CKD chronic kidney disease, Dis. disease. Source data are provided as a Source Data file.

**Table 2 | Descriptive statistics and evidence grading of the included meta-analyses of oral interventions**

| | Dental Treatment | Endodontic Treatment | Mandibular advancement | Periodontal Treatment |
|---|---|---|---|---|
| Number of meta-analyses | 4 | 1 | 2 | 172 |
| Number of studies, median (min-max) | 3.5 (2–15) | 3 | 10 (10–10) | 5 (2–25) |
| Number of participants, median (min-max) | 899 (426–2,163) | 100,701 | 450 (400–500) | 475.5 (46–7,335) |
| **Meta-analytical criterion, n (%)** | | | | |
| $P$ value $<10^{-6}$, n (%) | 0 (0) | 1 (100.0) | 1 (50.0) | 32 (18.6) |
| $P$ value $<10^{-3}$, n (%) | 0 (0) | 0 (0) | 0 (0) | 23 (13.4) |
| $P$ value $<0.05$, n (%) | 0 (0) | 0 (0) | 0 (0) | 29 (16.9) |
| $I^2 >50\%$, n (%) | 1 (25.0) | 0 (0) | 2 (100.0) | 98 (57.0) |
| $I^2 \leq 25\%$, n (%) | 2 (50.0) | 0 (0) | 0 (0.0) | 57 (33.1) |
| **Overall grading, n (%)** | | | | |
| Not significant | 4 (100.0) | 0 (0) | 1 (50.0) | 89 (51.7) |
| Weak | 0 (0) | 0 (0) | 1 (50.0) | 68 (39.5) |
| Suggestive | 0 (0) | 0 (0) | 0 (0) | 3 (1.7) |
| Highly suggestive | 0 (0) | 0 (0) | 0 (0) | 11 (6.4) |
| Strong | 0 (0) | 1 (100.0) | 0 (0) | 1 (0.6) |

Periodontitis accounted for the majority of the associations with NCDs and markers, followed by tooth loss and edentulism. The rise of Periodontal Medicine as an independent field in periodontal research can account for the relatively significant portion of such studies on periodontology[13]. While the mechanisms within this association have been progressively studied and understood, this bulk of knowledge has expanded to other areas, such as Peri-implantology and Endodontics. Primarily, the pathophysiology of periodontitis seems to be similar to the one of peri-implantitis and apical periodontitis for endodontic reasons, so it will not be surprising, shortly soon, for studies on the intersection of these pathologies with systemic diseases to draw a parallel with periodontitis. Concerning tooth loss and edentulism, both are mainly clinical endpoints of mainly periodontal diseases, and to a minor extent of dental caries[14,15].

Through methodological analysis and meta-analytic evidence, these results may have some degree of impact by biases. Only 10.3% of meta-analyses ($n = 30$) were conducted with high/moderate quality according to AMSTAR 2. While some of the observed issues may have a residual impact on the meta-analyses (such as, reporting funding from the included studies, accounting risk of bias on interpretation and discussion of the results, and defining the protocol a priori), others may have adverse effects to the consistency of the results (lack of a comprehensive search, absence of duplicate data search and extraction, and list of excluded studies with respective reason). Furthermore, from the statistical point-of-view, the included systematic reviews had, on average, relatively few studies (median=5) and few participants included (median = 996). Nevertheless, more than 55% of the included comparisons reported statistically significant results, with a substantial number, about 41.0% presenting a lower $P$-value threshold ($P < 10^{-6}$, $n = 351$). Around 60.5% ($n = 518$) showed high heterogeneity ($I^2 > 50\%$), and only 26.2% had low heterogeneity ($I^2 \leq 25\%$). FSN metrics results indicate a relatively strong consistency of the provided evidence, suggesting that most evidence is unlikely to change. For this reason, future research in the oral-systemic health intersection shall seek to strengthen compliance with established guidelines. Additionally, efforts should be made to increment the number of intervention trials to establish more conclusive inferences.

| | Improvement | | | | Non-significant | Worsening | | | |
|---|---|---|---|---|---|---|---|---|---|
| | Strong | Highly suggestive | Suggestive | Weak | | Weak | Suggestive | Highly suggestive | Strong |
| **Periodontal treatment** | | | | | | | | | |
| Adverse Pregnancy Outcomes | 0 | 3 | 3 | 5 | 31 | 0 | 0 | 0 | 0 |
| Blood Disorders | 0 | 0 | 0 | 1 | 1 | 0 | 0 | 0 | 0 |
| Cardiovascular Diseases | 0 | 1 | 0 | 3 | 6 | 0 | 0 | 0 | 0 |
| Chronic Kidney Disease | 0 | 0 | 0 | 1 | 0 | 0 | 0 | 0 | 0 |
| Gastrointestinal tract | 0 | 0 | 0 | 6 | 0 | 0 | 0 | 0 | 0 |
| Metabolic Disorders | 0 | 3 | 0 | 32 | 39 | 0 | 0 | 0 | 0 |
| Respiratory diseases | 0 | 1 | 0 | 0 | 0 | 0 | 0 | 0 | 0 |
| Rheumatic diseases | 0 | 0 | 0 | 10 | 1 | 0 | 0 | 0 | 0 |
| Systemic Inflammation | 1 | 3 | 0 | 10 | 11 | 0 | 0 | 0 | 0 |
| **Endodontic treatment** | | | | | | | | | |
| Cardiovascular Diseases | 1 | 0 | 0 | 0 | 0 | 0 | 0 | 0 | 0 |
| **Endodontic treatment** | | | | | | | | | |
| Cardiovascular Diseases | 0 | 0 | 0 | 0 | 2 | 0 | 0 | 0 | 0 |
| Mortality | 0 | 0 | 0 | 0 | 2 | 0 | 0 | 0 | 0 |
| **Mandibular advancement** | | | | | | | | | |
| Obstructive Sleep Apnea | 0 | 0 | 0 | 1 | 1 | 0 | 0 | 0 | 0 |

**Fig. 6 | Diagram showing results from the umbrella review grading the evidence of the effect of oral treatments (periodontal treatment, endodontic treatment, dental treatments or mandibular advancement) on NCDs.** The right side displays associations that increase the risk for the respective systemic NCD (in red), whereas the left side shows associations that reduce the risk (in green). APO adverse pregnancy outcomes, CVDs cardiovascular diseases, CKD chronic kidney disease. Source data are provided as a Source Data file.

These results convey a sufficient knowledge of hypothetical observational associations that shall not be ignored, although there is still insufficient understanding of the conceivable causal role of oral diseases on systemic diseases, particularly, and vice-versa. In this regard, public health data has shown that preventative actions are effective both in preventing oral diseases as well as increasing quality of life[16]. Still, in line with these results, research on intervention on oral diseases and their systemic impact is still unfavorable and is of paramount importance to be included in the international research agenda. The access to oral care and inequity in oral health and disease worldwide, poorly addressed in the included systematic reviews, also requires attention. While most of this evidence derives from research from developed countries, contributions from developing countries are still unrepresentative. While this is slowly being normalized, with the substantial increase in research by developing countries, these results should reflect the importance of access to oral health care, preventive and curative measures, and the potential impact on systemic health it may have. In parallel, many of the systemic NCDs that showed strong evidence of association are highly prevalent, show disproportional incidence rate and an exacerbated impact in underdeveloped nations[17,18].

Oral diseases might have an impact on systemic health via multiple pathways[9]. The oral microbiome, its byproducts and their interaction with the host immune system have been identified as the major players in this causal association. The largest body of evidence currently available focus on periodontitis as bacteremia and systemic inflammation represent plausible mechanisms of causality for the contribution of periodontitis to the pathogenesis of multiple diseases[9]. Chronic inflammation coupled with a richly vascularized periodontium leads to ulceration of the oral epithelial barrier, and consequently, greater access for pathogenic microbes and their products to the bloodstream[19]. Generally, the chronic systemic distribution of oral bacteria-derived products converges at the point of an altered state of immunity, achieved either through subversion of host defences, or prolonged and/or enhanced inflammatory responses. Low-grade systemic inflammation has been linked to the development of a wide spectrum of NCDs such as cardiometabolic, neurodegenerative, rheumatic and neoplastic conditions[20]. The increase in acute phase reactants and inflammatory cytokines could activate immune cells such as circulating monocytes with a subsequent vascular inflammation[21]. Also, chronic inflammation can cause lipid and lipoprotein metabolism changes leading to proatherogenic lipoproteins setting[22]. In addition, systemic inflammation is associated with glucose intolerance and insulin resistance contributing to increased risk of type 2 diabetes mellitus and other metabolic disorders[23]. The action of oral bacteria such as *Porphyromonas gingivalis* (*P. gingivalis*), *Aggregatibacter actinomycetemcomitans* (*A. actinomycetemcomitans*) and *Fusobacterium nucleatum* (*F. nucleatum*) has been linked to the development of rheumatic conditions and certain types of cancer. *P. gingivalis* and *A. actinomycetemcomitans* have shown the ability to trigger the production of anti-citrullinated protein antibodies (ACPAs)[24–26], a classic feature detected in rheumatoid arthritis. *F. nucleatum* in in vitro and animal models can stimulate the growth and migration of colorectal cancer (CRC) cells[27–30] and accelerates tumor growth and metastatic progression in breast cancer[31]. Furthermore, immune cells, specifically lymphocytes and myeloid cells, can be primed during the interaction with inflamed oral sites and have a detrimental effect at distant mucosal sites[32]. The effect of the treatment of periodontitis on several markers of systemic inflammation and disease activity, surrogate markers of cardiovascular health and metabolic control has corroborated the hypothesis of a causal association with NCDs[33–35].

Other non-biological but shared pathways that mediate this association concern nutrition and social determinants, both recognized as relevant modifiable factors in this intersection[36,37]. Compromised oral health decisively affects dietary pattens, with a conceivable impact on NCDs where nutrition is a key contributor, such as metabolic disorders (e.g., diabetes mellitus or obesity)[36]. Nutritional imbalances also shape the human body throughout life, including the impact on the development of oral structures (for instance, dysvitaminosis) or modeling gut microbiome[38] and inflammation pathways[39]. In addition, social and behavioral determinants of oral health have been emerging due to the higher incidence of oral diseases in regions characterized by low income, low educational attainment and deprived socioeconomic status[1,2]. These determinants are directly influenced by the ability to access good oral health care and appropriate health behaviors, and addressing them is detrimental to reduce health disparities[1,2,37].

The protocol of the present umbrella review was developed a priori and registered in PROSPERO, contributing to the robustness of its analyses and results, transparency and mitigation of errors. We did not limit this review to assessing the methodological quality but also focused on the degree of evidence of meta-analytic estimates based on

the strategy defined by Papadimitriou et al.[40]. As argued, despite the existence of other methods for rating evidence quality, such as GRADE[41], this approach focuses on objective and tangible criteria. Additionally, methodological quality via the AMSTAR 2 allowed a comprehensive view from both randomized and non-randomized studies[42], and goes beyond the statistical sphere of meta-analyses. As well, the use of FSN allowed us to learn that 97.5% of the significant evidence (from suggestive to strong) is unlikely to change even in the possibility of future research, thus allowing robust conclusions to be implemented in public health agenda and policymakers. Nevertheless, these indications, without limiting research to the associations that achieved strong statistical significance, aim to foster the interest in filling in the existing science and knowledge gaps, which are still many given the relative youth of the intersection of oral health with health in general. In addition, this notion of completed science is unreal, as there are situations of strong associations with a short bulk of research carried out and highly suggestive or suggestive associations with a remarkable volume of research.

However, this review presents important shortcomings worth discussing. The current umbrella review is heavily based on meta-analyses of observational studies and with a low percentage of longitudinal prospective studies and randomized trials. Hence, the overall view of these results relies more on non-inferential evidence than definitive causal assumptions. Still, we highlight some of the observational studies included have a considerable number of participants, and some, although scarce, have a prospective design and come from insurance databases with a population-level data source, such as the Taiwan's National Health Insurance Research Database[43]. This sort of databases represents an opportunity to progress big data analysis by combining multiple groups of variables (physical, laboratorial, clinical or sociodemographic, among others). Nonetheless, the validity of the diagnosis codification codes (mostly based on international classification of disease 9 clinical modifications [ICD-9-CM] and ICD-10) is still controversial, because while the reliability of ICD-9-CM is somehow well established, ICD-10 codes are yet unvalidated[43].

Regarding the observational nature of the majority of studies included in the meta-analyses, there is a large variability in the measurement or diagnosis in commonly seen oral diseases, such as dental caries or periodontal disease. This heterogeneity may constitute a source of bias and may lead to under- or overestimated meta-analytical results. On the one hand, dental caries was mainly reported through the Decayed, Missing, and Filled Teeth (DMFT) index, the most commonly used epidemiological index for assessing dental caries[44]. While this index provides a complete view of the history of caries, it overestimates caries experience by attributing the cause of all missing teeth to caries. In this case, assigning the maximum or minimum value is likely to over- or underestimate inequalities, respectively, and ignoring the 'M' component would omit possibly a major inequality component of this index[44]. On the other hand, the periodontal clinical measures, particularly clinical attachment loss (CAL) and periodontal probing depth (PPD), are often provided as a global sum of the patient's mouth over thresholds of pathological PPD or CAL within the whole mouth[45]. Likewise, the variability of periodontitis and gingivitis case definitions is also a possible source of bias. Recently a joint consensus case definition for periodontitis was proposed[46], and the variation from previous case definitions was attested using graphical representations[47]. The same is worth noting for the various systemic diseases herein presented, since most have suffered changes in their diagnosis or disease staging, contributing to high heterogeneity and possible unstable meta-analytical statistical power and significance.

One additional shortcoming is the fact that most meta-analyses produced association estimates based on unmeasured confounding[48,49]. In the studies included in this review, most explored the impact of confounding factors through subgroup meta-analyses or meta-regression. These methods are limited and do not minimize the potential confounding bias in meta-analysis[49], and the assessment of confounder adjustment strategies shall be considered in the future. To our view, this can be achieved using reported effect sizes already adjusted for confounding factors. Nevertheless, the heterogeneity and variability of confounding factors reported in studies may limit this ultimate challenge, and for this reason an overall framework to improve the reporting of adjusted estimates is recommended. Thus, future systematic reviews and, to a greater extent primary studies, shall improve data reporting by providing effect sizes adjusted for significant confounding factors and therefore increasing the consistency of results and preventing residual confounding.

It is important to note that data independence is an elementary condition for valid statistical analysis. For instance, when data from the same patients' cohort is published in different papers, care should be taken to avoid data duplicity during data pooling. Ideally, samples from the same cohort published in different articles should be grouped under one study name and have their data reported together, based on the original characteristics of the sample. This will preclude the multiple use of identical data from the same studies for estimating an association. Therefore, the interpretation of meta-analysis where identical/overlapping data could be identified should be interpreted with caution, as the pooled $P$ values reported in the original review are not valid (this may also affect the interpretation and conclusions of a review). In this umbrella review, data dependence was residual (3.74% of the meta-analyses), and such estimates were matched by low-graded methodological quality and weak evidence strength.

In what meta-analyses from intervention studies are concerned, the short follow-up studies in overall oral research are also a major restraint, as previously discussed[13,50]. As a result of the knowledge obtained from the impact of periodontal therapies on systemic inflammation[35] and glycemia[51], these outcomes may occur several months postintervention, thus oral research community shall expand the follow-up longer to allow a more convincing and lasting outlook.

## Implications for practice and research

The elevated number of meta-analyses included roots the notion that the links between oral diseases and systemic NCDs have been a topic of growing interest. While the overall evidence consistency is unfavorable due to poor meta-analytic or methodological reasons, these results substantiated significant associations with NCDs that are prominently prevalent, such as diabetes mellitus, CVDs, rheumatic or neurodegenerative conditions. This does not mean that other robust associations cannot endure, yet the current body of evidence does not support such inferences. Until the level of evidence becomes clearer and the linking mechanisms fully understood, agencies dedicated to public health (such as the WHO) have alluded to the importance of prevention (primary, secondary and tertiary) and modeling impactful modifiable factors. Enhancing prevention attitudes and programs and addressing social determinants of oral health have always contributed to major shifts in oral health standards, such as water fluoridization or food sugar content[1]. With this in mind, healthy systems and oral care providers shall focus on at-risk patients through preventive programs for early detection, oral health literacy promotion, using reliable diagnostic approaches and implementing adequate treatments. Furthermore, all interesting parties shall contribute to a global view of health with the mouth as an integral part of the body, and this will foster a growing multidisciplinary care of the patient within its specifics and particularities. Enriching the curricula of medical and dental health courses is an initiative with conceivable impact in the long-term, along with dedicated training towards this unique one health vision, and may expand the observant radar for the impact of oral status on systemic health and vice-versa. These results also enhance the available information with evidence grading maps that may enhance to the engagement of policymakers towards increasingly refined programs embodied with the most current evidence.

## Methods

### Protocol and reporting

All authors defined the protocol a priori, and details of the protocol for this systematic review were registered on PROSPERO (ID: CRD42022300740) and can be accessed at https://www.crd.york.ac.uk/prospero/display_record.php?ID=CRD42022300740. The umbrella review is reported following the PRISMA guideline[52] and a PRISMA checklist is included (Supplementary Data 6).

### Study selection

For this umbrella review, five electronic databases (PubMed, Cochrane Database of Systematic Reviews, EMBASE, Web of Science, and LILACS) were searched up to December 2021. We merged keywords and subject headings appropriately for each database using the following syntax: (periodontal disease[MeSH] OR oral health[MeSH] OR dental Caries[MeSH] OR "oral manifestations") AND ("systematic review" OR "meta-analysis" OR "meta-analysis") (this search syntax represent a post-hoc deviation from the protocol to avoid the disregard of meta-analyses that were part of a systematic review but whose mention to "systematic review" term in the title or on the abstract were not present). In addition, grey literature was searched via http://www.opengrey.eu. Additional relevant literature was included after a manual search of the reference lists of the final included articles. The electronic database search was carried out by two independent authors (J.B. and V.M.), and the final decision for inclusion was made according to the following criteria: (1) systematic reviews with meta-analysis; (2) results from human studies; (3) assessing the association between oral and systemic conditions. There were no restrictions regarding the year or language of publication. As such, exclusion criteria were as follows: (1) systematic reviews without meta-analysis were excluded as it prevented the quantification of the meta-analytical quality of the estimates; (2) systematic reviews reporting binary results without controls; (3) systematic reviews failing to provide meta-analytic estimates and heterogeneity results; (4) systematic reviews of systematic reviews (umbrella reviews). Additional post-hoc decisions about exclusion of studies were set, regarding some specificities found during studies inclusion: (a) commentaries, abstracts, letter to the editors or consensus; (b) systematic reviews restricted to studies of a particular country; (c) lack of appropriate clinical measures; (d) secondary analysis from data sourced from a previous systematic review; (e) unsuitable inclusion criteria; and, (f) including animal studies in the meta-analysis.

### Data extraction

We prepared a predefined table to extract the necessary data from each eligible systematic review, including: study identification (authors and year), number of studies included in the meta-analysis, type and number of studies included, oral condition(s) being assessed, systemic condition(s) being assessed, methodological quality tool used, effect size and 95% CI, funding information. From each eligible systematic review, three independent researchers (J.B., V.M., J.V.) extracted information and all disagreements were resolved through discussion with a fourth reviewer (J.J.M.). The agreement between the examiners was considered excellent (0.88, 95% CI: 0.86–0.90).

### Methodological quality appraisal

The included systematic reviews were independently assessed by two examiners (J.B. and V.M.) using the AMSTAR 2[42]. In this sense, systematic reviews are categorized as: High (Zero or one non-critical weakness); Moderate (More than one non-critical weakness); Low (One critical flaw with or without non-critical weaknesses); and Critically Low (More than one critical flaw with or without non-critical weaknesses).

### Grading of the evidence

We graded meta-analyses following a previously published methodology[40]. Significant associations were categorized into four evidence levels: strong, highly suggestive, suggestive, and weak evidence[40,53]. A category of strong evidence was attributed if all the following criteria were met: >1000 cases included in the meta-analysis, a threshold that provides 80% power for hazard ratios $\geq 1.20$ ($\alpha = 0.05$)[40]; a $P$-value $\leq 10^{-6}$ of statistical significance in valid meta-analysis[54–56]; heterogeneity ($I^2$) below 50%; the null value was excluded by the 95% prediction interval; and, no evidence of small study effects and excess significance bias. Highly suggestive evidence was set if: meta-analyses with >1000 cases; a random effects $P$-value $\leq 10^{-6}$, and the largest study in the meta-analysis was statistically significant. Suggestive evidence was defined if: meta-analyses with >1000 cases, random effects $P$-value $\leq 10^{-3}$[54–56] were categorized. If the latter conditions were not verified, the meta-analysis was classified as weak evidence.

### Calculation of FSN

In nominally statistically significant meta-analyses, we determined the number of future studies of average null effect and average weight needed to detect a non-statistically significant summary estimate by calculating Rosenberg's FSN[57]. We used the Meta-Essentials packages for binary (odds ratio, risk ratio, hazard ratio, incidence ratio or ratio of means) and continuous measures (mean difference, standardized mean difference or weighted mean difference)[58]. We then calculated the median and range for each evidence grade (strong, highly suggestive, suggestive and weak).

### Data handling and management

All data were collected in MS Office 365. Inferential statistical analyses were computed using R version 4.03.

### Reporting summary

Further information on research design is available in the Nature Portfolio Reporting Summary linked to this article.

## Data availability

All data generated and analyzed during this study are provided in the Supplementary Information. Source data are provided with this paper.

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

## Acknowledgements

This work is financed by national funds through the FCT—Foundation for Science and Technology, I.P., under the Project UIDB/04585/2020. The study sponsor had no role in the design and conduct of the study; collection, management, analysis and interpretation of the data; preparation, review or approval of the article; and decision to submit the article for publication.

## Author contributions

The study was conceived and designed by J.B. The data were acquired and collated by J.B., V.M., and J.V. and analyzed by J.B. and V.M. The manuscript was drafted by J.B. and revised critically for important intellectual content by J.B., P.M., J.V., L.P., M.O., Y.L., L.C., J.J.M. and V.M. All authors gave final approval of the version to be published and have contributed to the manuscript.

## Competing interests

The authors declare no competing interests.
