## [Peer Review File · Nature Communications]

An umbrella review of the evidence linking oral health and systemic noncommunicable diseasesREVIEWER COMMENTS

Reviewer #1 (Remarks to the Author):

This is an interesting and important umbrella review of the evidence linking oral health and systemic health. The authors have clearly undertaken a great deal of work in assessing the published literature and this paper has the potential to make a significant contribution to future research and policy.

The paper however needs to be revised in the following ways:

1. The authors need help in improving their English throughout the manuscript as in many sections the text is unclear or confusing.
2. The introduction needs to be sharper and provide a better justification for this umbrella review. A more specific aim for the review would be helpful and the fact that the association is bi-directional. The text in lines 70-73 reads more like a conclusion rather than introduction.
3. Why is the methods not presented before the results section? Is this in line with the journal guidance?
4. The results are very interesting and figures 2-5 are very clear. I could not find tables 1 and 2 nor figures 6 and 7.
5. The wide range in values for the FSN needs to be commented upon (see lines 192-198).
6. The discussion section raises some important points but further consideration is needed on mechanisms beyond inflammation such as other shared pathways as nutrition and social relationships.
7. The implications section needs to be reviewed and I would challenge the authors on the basis for their practice recommendations (see lines 370-373) What is the basis for these specific recommendations?

Reviewer #2 (Remarks to the Author):

The authors have done a huge work trying to sum up all findings linking oral infections to systemic conditions: 213 systematic reviews and 634 meta-analytic comparisons were included in this umbrella review. Although the work is potentially interesting, the presentation could be improved. Also, tables were missing.

Specific comments:

-“Oral conditions”, “oral disorders”, “oral diseases”; please use only one expression and add, what oral diseases are considered in the review.

-“Robust evidence supports the greater prevalence of oral conditions in people suffering from NCDs limiting the ability of oral self-care”; limiting the ability of oral self-care has not been shown in the analyses.

-The conclusions in the abstract should be rephrased: “Most evidence is unlikely to change which indicates a relatively robust consistency of the available body of evidence.”

-Table 1 presenting the summary descriptive characteristics and Table 2 exploring the impact of oral treatment on systemic health are not included in any of the files.

-“Figure 8. Diagram showing results from the umbrella review grading the evidence on diet and cancer

risk.” I do not understand the figure title.

-Figures 6 and 7 do not exist.

-I wonder whether figures 2 and 3 are correct: in Figure 2 mortality increases the risk of periodontitis.

Reviewer #3 (Remarks to the Author):

The authors present an overview of meta-analyses assessing the association between oral health with systemic non-communicable diseases. An assessment of the strength, validity and consistency of the relationships is also presented.

I have some items for the authors consideration, the majority are re the terminology used systematic review versus meta-analysis and with the presentation of the results, which I think requires some further consideration.

Line 40-41 and 70-71. It is unclear why NCDs and cancer are differentiated here. Cancer is an NCD.

Perhaps, “26 NCDs including four types of cancer”?

Line 42. “...significant NCDs..” Is this referring to statistically significant associations? If so, present accordingly.

Regarding study selection, it is unclear what the authors did with published “meta-analyses” i.e. meta-analyses that appear in publication that don’t lay claim to being part of a systematic review. These are relatively common. Were any encountered during the course of this work? If so, how were they handled?

Line 411. Agreement between reviewers is quantified. What is this statistic presented? (Kappa?) It should be identified.

Suppl. File 1. Reasons for exclusion should align directly to the review inclusion criteria. There are examples here that do not. Eg #64 Restricted to a particular country – was this a post-hoc decision/deviation from protocol?

The authors should be wary of their terminology In the first couple of paragraphs of the results. Meta-analyses are being referred to where I think “systematic reviews” or systematic reviews with meta-analyses” should be. Eligibility and quality has been established for the reviews, yet your assessments of the results – the grading - is based on individual, study-outcome, meta-analyses, not the systematic review they are a part of. It is important to differentiate in your results where the reader should transition from discussion of a review versus a meta-analysis. Indeed the 213 systematic reviews presented 634 meta-analyses.

Line 93 I don’t think PRISMA, MOOSE and/or Cochrane is a legitimate comparison here as these are guidelines with different intent. PRISMA and MOOSE are standards for reporting a systematic review, Cochrane present standards of conduct (and recommend adhering to PRISMA reporting standards). Table 1 has not been provided for review.

Figure 2, second row. Should read “blood disorders” rather than “blood levels”? The figure presents periodontal diseases as an outcome (~40% of the include meta-analyses – line 101), yet the places the red and green as increased and decreased risk respectively of the systemic condition? I am not familiar with the construction of these diagrams, yet I would expect it to present the risk of the outcome, not the

exposure? The association appears to have been flipped in this figure? While there is evidence for a bidirectional association between oral and systemic conditions, I don't think that 'flip' can be made in this way and the evidence should be presented directly as it was assessed/analysed in the reviews and analyses.

On page 6, authors should provide reference to which figures the reader should refer. The figure labelling isn't entirely informative (which periodontal disease for example) and it is difficult to navigate. For example, line 114-117 appears to correspond to Fig 4, Oral condition as an exposure, yet there is no corresponding data in the figure with tooth loss as an exposure and cognitive impairment for example? The order of presentation of figures should correspond to their order in the text also.

Table 2 has not been provided for review.

Line 157 Considering the aims – establishing relationship between oral and systemic health, I am unclear how/why oral treatment is being considered here or how the link has been made between treatment for an oral condition and the outcome. Were these reported/recorded as adverse effects during or following the intervention? How has the treatment been differentiated from the oral condition which may be associated with the outcome? Regardless, this section appears to be deviating from the original intent of the work as described.

Figure 8 has not been provided.

Suppl data 2 and 3 are not referred to in the manuscript.

Line 203 Should be 213 systematic reviews.

Line 248-250. Has the median number of studies and participants been established for the included reviews, or the included meta-analyses? The latter would seem the more important considering the point being made re statistical assessment. A meta-analysis may include fewer studies than the parent review, or indeed, include discrete cohorts from one study that may impact measures included here that have been used for grading of the evidence.

Line 322-324 warrants a citation.

REVIEWER COMMENTS

Reviewer #1 (Remarks to the Author):

This is an interesting and important umbrella review of the evidence linking oral health and systemic health. The authors have clearly undertaken a great deal of work in assessing the published literature and this paper has the potential to make a significant contribution to future research and policy.

Our answer: Thank you for your remarks. We have addressed all and the point-by-point responses are below.

The paper however needs to be revised in the following ways:

1. The authors need help in improving their English throughout the manuscript as in many sections the text is unclear or confusing.

Our answer: We appreciate this remark. The manuscript was revised by a native English speaker in order to improve language fluency.

2. The introduction needs to be sharper and provide a better justification for this umbrella review. A more specific aim for the review would be helpful and the fact that the association is bi-directional. The text in lines 70-73 reads more like a conclusion rather than introduction.

Our answer: We followed this reviewer suggestion of providing a better justification for this umbrella review by rephrasing the aims of the study. Now reads as follows:

“For this reason, we aimed to perform an umbrella review to overlook the robustness of the meta-analytic estimates linking oral and systemic diseases and its bidirectional association. We additionally aimed to explore whether future research will likely transform the inferences from existing significant meta-analyses.”

Regarding lines 70-73, indeed this piece of text was designed more like a conclusive statement of the results of the work, as its structure was based on publications from Nature Communications. We took the chance to state below some recent examples that used such writing approach:

1. <https://www.nature.com/articles/s41467-022-30430-4>
2. <https://www.nature.com/articles/s41467-022-30357-w>
3. <https://www.nature.com/articles/s41467-022-30293-9>

3. Why is the methods not presented before the results section? Is this in line with the journal guidance?

Our answer: Methods were not presented before the Results section considering and in line with the 'Guide for authors' in Nature Communications (<https://www.nature.com/ncomms/submit/article>). In this journal, "The main text of an Article should begin with a section headed Introduction (...), followed by sections headed Results, Discussion (...) and Methods (...)".

3. The results are very interesting and figures 2-5 are very clear. I could not find tables 1 and 2 nor figures 6 and 7.

Our answer: We apologize for this informatic lapse. We have provided both Tables 1 and 2 but for some reason they were not available in the submission platform. Also, Figure 8 should have been named Figure 6. We have thoroughly revised the order of the figures and their availability in the document.

4. The wide range in values for the FSN needs to be commented upon (see lines 192-198).

Our answer: Following your suggestion, we commented on the wide range of values by adding a new sentence, that now reads as follows: "Among the 294 meta-analyses that achieved from suggestive to strong evidence, the median fail-safe number (FSN) was 51 with a notable wide range of values (range: 1 to 87,973)."

5. The discussion section raises some important points but further consideration is needed on mechanisms beyond inflammation such as other shared pathways as nutrition and social relationships.

Our answer: We considered this commentary as extremely valid. Having this said, we added a new paragraph addressing the importance of these shared pathways, that now reads as follows:

"Other non-biological but shared pathways that mediate this association concern nutrition and social determinants, both recognized as relevant modifiable factors in this intersection^{34,35}. Compromised oral health decisively affects dietary patterns, with a conceivable impact on NCDs where nutrition is a key contributor, such as metabolic disorders (e.g., diabetes mellitus or obesity)³⁴. Nutritional imbalances also shape the human body throughout life, including the impact on the development of oral structures (for instance, dysviteaminosis) or modulating gut microbiome³⁶ and inflammation pathways³⁷. In addition, social and behavioral determinants of oral health have been emerging due to the higher incidence of oral diseases in regions characterized by low-income, low educational attainment and deprived socioeconomic status^{1,2}. These determinants are directly influenced by the ability to access good oral health care and appropriate health behaviors, and addressing them is detrimental to reduce health disparities^{1,2,35}."

7. The implications section needs to be reviewed and I would challenge the authors on the basis for their practice recommendations (see lines 370-373) What is the basis for these specific recommendations?

Our answer: We have revised this section according to your remark. These recommendations were based on the promotion from specialized agencies dedicated to public health (such as the WHO) towards the importance of modeling impactful modifiable factors. Addressing such determinants have resulted in major shifts in oral health standards, such as water fluoridation or food sugar content. For this reason, primary prevention and improving oral health literacy have been proposed to address the elevated incidence and burden of oral diseases, and with these in mind we proposed such recommendations. To better clarify this, the following changes were made as follows:

“Until the level of evidence becomes clearer and the linking mechanisms fully understood, agencies dedicated to public health (such as the WHO) have alluded to the importance of prevention (primary, secondary and tertiary) and modeling impactful modifiable factors. Enhancing prevention attitudes and programs, and addressing social determinants of oral health have always contributed to major shifts in oral health standards, such as water fluoridation or food sugar content 1. With this in mind, healthy systems and oral care providers shall focus on at-risk patients through preventive programs for early detection, oral health literacy promotion, using reliable diagnostic approaches and implementing adequate treatments.”

Reviewer #2 (Remarks to the Author):

The authors have done a huge work trying to sum up all findings linking oral infections to systemic conditions: 213 systematic reviews and 634 meta-analytic comparisons were included in this umbrella review. Although the work is potentially interesting, the presentation could be improved. Also, tables were missing.

Our answer: Thank you for your remarks. We have addressed all and the point-by-point responses are below.

Specific comments:

-“Oral conditions”, “oral disorders”, “oral diseases”; please use only one expression and add, what oral diseases are considered in the review.

Our answer: Following this valid recommendation we have standardized all those three terms in one common expression: “oral diseases”, which is now used throughout the manuscript (we have not added all marked changes here to maintain the readability of the letter). As well, we added a brief summary on what we considered as oral diseases in the review, as follows: “Oral diseases are chronic and progressive conditions that affect the health of teeth and mouth” (Page 3).

-“Robust evidence supports the greater prevalence of oral conditions in people suffering from NCDs limiting the ability of oral self-care”; limiting the ability of oral self-care has not been shown in the analyses.

Our answer: We appreciate this observation, and this sentence did not aim to summarize a result (and we acknowledge the reviewer's observation which, albeit correct, was not analyzed in this work). This sentence served as an introductory text to the research topic prior to the presentation of the study aim, considering the unstructured format proposed by this Journal.

-The conclusions in the abstract should be rephrased: “Most evidence is unlikely to change which indicates a relatively robust consistency of the available body of evidence.”

Our answer: We have rephrased accordingly, and now reads: “According to fail-safe number statistics, the evidence levels are unlikely to change in the future, indicating a fairly robust consistency.”

-Table 1 presenting the summary descriptive characteristics and Table 2 exploring the impact of oral treatment on systemic health are not included in any of the files.

Our answer: We apologize for this informatic lapse. We have provided both Tables 1 and 2. For some reason they were not available in the submission platform.

-“Figure 8. Diagram showing results from the umbrella review grading the evidence on diet and cancer risk.” I do not understand the figure title.

Our answer: This was a typo. We have rephrased to “Figure 6”. Diagram showing results from the umbrella review grading the evidence of the effect of oral treatments (periodontal treatment, endodontic treatment, or dental treatments) on NCDs.”

-Figures 6 and 7 do not exist.

Our answer: We apologize for this informatic lapse. For some reason, Figure 8 (present on page 10) should have been named Figure 6. We have thoroughly revised the order of the figures and their availability in the document.

-I wonder whether figures 2 and 3 are correct: in Figure 2 mortality increases the risk of periodontitis.

Our answer: We have carefully double-checked Figures 2 and 3, that now are Figures 4 and 5, according to the reordering of figures. You are totally right; the captions were changed by mistake. We corrected this accordingly in the manuscript.

Reviewer #3 (Remarks to the Author):

The authors present an overview of meta-analyses assessing the association between oral health with systemic non-communicable diseases. An assessment of the strength, validity and consistency of the relationships is also presented.

I have some items for the authors consideration, the majority are re the terminology used systematic review versus meta-analysis and with the presentation of the results, which I think requires some further consideration.

Our answer: Thank you for your remarks. We have addressed all and the point-by-point responses are below.

Line 40-41 and 70-71. It is unclear why NCDs and cancer are differentiated here. Cancer is an NCD. Perhaps, "26 NCDs including four types of cancer"?

Our answer: We appreciate this remark very much. We have rephrased accordingly to both in lines 40-41 and 70-71.

Line 42. "...significant NCDs.." Is this referring to statistically significant associations? If so, present accordingly.

Our answer: We were not referring to statistically significant associations but rather the strong evidence obtained. To make this clear, we have rephrased to "(...) those NCDs (...)" .

Regarding study selection, it is unclear what the authors did with published "meta-analyses" i.e. meta-analyses that appear in publication that don't lay claim to being part of a systematic review. These are relatively common. Were any encountered during the course of this work? If so, how were they handled?

Our answer: Thank you for this commentary. In fact, meta-analyses that have not laid claim to being part of a systematic review posed a methodological obstacle and potentially led to the possibility to be disregarded in the search process. After carefully reading your commentary, we were alerted to the possibility that some meta-analyses, that were part of a systematic review, may have been lost in the search process because not including the term "systematic review" on the title or on the abstract were not "found" by the databases' search engine. For this reason, considering the amount of work and the depth of our work, we decided to re-run the search using the following search syntax (presented as a PubMed syntax, and adapted accordingly to the remaining databases):

(periodontal disease[MeSH] OR oral health[MeSH] OR dental Caries[MeSH] OR "oral manifestations") AND ("systematic review" OR "meta-analysis" OR "metaanalysis")

We have made the necessary alterations to the manuscript, highlighting this was a post-hoc decision and a deviation from the protocol: "(this search syntax represent a post-hoc deviation from the protocol to avoid the disregard of meta-analyses that were part of a systematic review but whose mention to "systematic review" term in the title or on the abstract were not present)".

Consequently, and considering the effort of updating this search we took the opportunity to revise the search until May 2022. Therefore, the final number of included systematic reviews and this umbrella review were updated accordingly. All results were updated accordingly, and whose results and conclusions have not been altered.

Line 411. Agreement between reviewers is quantified. What is this statistic presented? (Kappa?) It should be identified.

Our answer: We have identified the statistic presented as “Cohen's kappa score”.

Suppl. File 1. Reasons for exclusion should align directly to the review inclusion criteria. There are examples here that do not. Eg #64 Restricted to a particular country – was this a post-hoc decision/deviation from protocol?

Our answer: Indeed, this is a case of a post-hoc decision/deviation from protocol. According to Cochrane Handbook (Chapter 3), we have reported the additional exclusion criteria that were set up regarding some specificities found during studies inclusion. The added text now reads as follows: “Additional post-hoc decisions about exclusion of studies were set, regarding some specificities found during studies inclusion: (a) commentaries, abstracts, letter to the editors or consensus; (b) systematic reviews restricted to studies of a particular country; (c) lack of a systemic condition; (d) lack of appropriate clinical measures; (e) secondary analysis from data from a previous systematic review; (f) unsuitable inclusion criteria; and, (g) including animal studies in the meta-analysis.” (Pages 20-21).

The authors should be wary of their terminology In the first couple of paragraphs of the results. Meta-analyses are being referred to where I think “systematic reviews” or systematic reviews with meta-analyses” should be. Eligibility and quality has been established for the reviews, yet your assessments of the results – the grading - is based on individual, study-outcome, meta-analyses, not the systematic review they are a part of. It is important to differentiate in your results where the reader should transition from discussion of a review versus a meta-analysis. Indeed the 213 systematic reviews presented 634 meta-analyses.

Our answer: We completely agree with this remark. We have followed your recommendation and have differentiating, in the results, systematic reviews until we discuss meta-analyses per se, to allow a smoother transition from discussion of a review versus a meta-analysis.

Line 93 I don't think PRISMA, MOOSE and/or Cochrane is a legitimate comparison here as these are guidelines with different intent. PRISMA and MOOSE are standards for reporting a systematic review, Cochrane present standards of conduct (and recommend adhering to PRISMA reporting standards).

Our answer: We agree with this remark. We have rewritten Lines 90-93 accordingly, removing the reference to studies that have reported using Cochrane's standards of conduct.

Table 1 has not been provided for review.

Our answer: We apologize for this informatic lapse. We have provided both Tables 1 and 2, for some reason they were not available in the submission platform.

Figure 2, second row. Should read “blood disorders” rather than “blood levels”? The figure presents periodontal diseases as an outcome (~40% of the include meta-analyses – line 101), yet the places the red and green as increased and decreased risk respectively of the systemic

condition? I am not familiar with the construction of these diagrams, yet I would expect it to present the risk of the outcome, not the exposure? The association appears to have been flipped in this figure? While there is evidence for a bidirectional association between oral and systemic conditions, I don't think that 'flip' can be made in this way and the evidence should be presented directly as it was assessed/analysed in the reviews and analyses.

Our answer: We appreciate this observation very much, and we acknowledge its importance. In fact, we presented figures in this way considering the nature of each meta-analysis, that is, there were two groups of meta-analyses: 1) those who considered the "oral disease" as an outcome of a NCD (in other words, the likelihood of people suffering from that NCD compared to healthy counterparts); 2) and, those who considered the "oral disease" and a potential risk for a NCD being present (as an "exposure"). Therefore, and considering the completely different nature of those, we could not integrate them into a single figure, and therefore we had to present different grading maps.

On page 6, authors should provide reference to which figures the reader should refer. The figure labelling isn't entirely informative (which periodontal disease for example) and it is difficult to navigate. For example, line 114-117 appears to correspond to Fig 4, Oral condition as an exposure, yet there is no corresponding data in the figure with tooth loss as an exposure and cognitive impairment for example? The order of presentation of figures should correspond to their order in the text also.

Our answer: Thank you for highlighting this point. We agree that providing reference to which figures the reader should refer would contribute to a clear interpretation from the reader. We thus added this piece of information along the subsections **Grading of the evidence from oral diseases** (Page 5) and **Grading of the evidence from the impact of oral treatments** (Page 9). We also reordered the figures to correspond to their order in the text. We organized figures to include for each oral disease both possibilities (as an exposure or an outcome) to make them clearer to interpret.

Table 2 has not been provided for review.

Our answer: We apologize for this informatic lapse. We have provided both Tables 1 and 2, for some reason they were not available in the submission platform.

Line 157 Considering the aims – establishing relationship between oral and systemic health, I am unclear how/why oral treatment is being considered here or how the link has been made between treatment for an oral condition and the outcome. Were these reported/recorded as adverse effects during or following the intervention? How has the treatment been differentiated from the oral condition which may be associated with the outcome? Regardless, this section appears to be deviating from the original intent of the work as described.

Our answer: Once again, we appreciate this remark. Systematic reviews addressing the effect of oral treatments on a particular NCD reflected the impact of treating an existing oral disease and how it could affect a particular marker/sign of a disease. As an example, several systematic reviews explored how periodontal therapy (based on mechanical debridement) could reduce the risk for adverse pregnancy outcomes if employed before or in the course of pregnancy. In such cases, the oral disease had already been diagnosed and this sort of systematic reviews intended to go beyond the association, that is, they were developed to confirm the causality between the treatment of a particular oral disease and an NCD. To our view, this group of systematic reviews

are still within the scope of the systematic review, since it confirms this causality based on the treatment of oral disease.

Figure 8 has not been provided.

Our answer: We apologize for this informatic lapse. For some reason, Figure 8 (present on page 10) should have been named Figure 6. We have thoroughly revised the order of the figures and its availability in the document.

Suppl data 2 and 3 are not referred to in the manuscript.

Our answer: We appreciate this commentary. We have double-checked and Suppl data 2 was mentioned in lines 82-83 and line 100. As well sup data 3 was mentioned in line 90. Thank you for your thoughtfulness.

Line 203 Should be 213 systematic reviews.

Our answer: We have rephrased to the new updated number (294), yet we acknowledge that originally it should have been 213.

Line 248-250. Has the median number of studies and participants been established for the included reviews, or the included meta-analyses? The latter would seem the more important considering the point being made re statistical assessment. A meta-analysis may include fewer studies than the parent review, or indeed, include discrete cohorts from one study that may impact measures included here that have been used for grading of the evidence.

Our answer: The median number of studies has been established for the included meta-analyses, and we agree with this reviewer that is indeed more important. As well, the applied method was based on Papadimitriou et al. (2021).

Line 322-324 warrants a citation.

Our answer: We appreciate this suggestion. This sentence (lines 322-324) is an interpretation of our own results. We hope to have clarified this point, yet we are totally available to make further arrangements if you still find them necessary.

REFERENCES USED

Papadimitriou, N. et al. An umbrella review of the evidence associating diet and cancer risk at 11 anatomical sites. Nat. Commun. 12, 4579 (2021).

REVIEWER COMMENTS

Reviewer #1 (Remarks to the Author):

The authors have now addressed the reviewers comments and the paper is much improved as a consequence. The quality of the English is however still very mixed and needs a comprehensive edit.

Reviewer #2 (Remarks to the Author):

Thank you for the carefully done revision of the manuscript. I have only minor further suggestions:

-This statement requires a reference: "Concerning tooth loss and edentulism, both are mainly clinical endpoints of mainly periodontal diseases, and to a minor extent of dental caries." The reasons for tooth loss are associated with age.

-Please, correct this and other species names: Porphyromonas Gingivalis -> Porphyromonas gingivalis

Reviewer #4 (Remarks to the Author):

1. This overview of systematic reviews (SRs) included 294 systematic reviews, with 856 meta-analytic comparisons, regarding associations between oral conditions and systemic diseases (NCDs). The study found strong or suggestive associations between many NCDs (including some cancers) and oral health conditions.

2. I feel that authors' response to comments from Reviewer #3 are generally satisfactory, but not sure about the response concerning Fig 2. If my understanding is correct, the authors considered the included meta-analyses as two categories: (1) oral conditions as outcomes or (2) as exposures. However, I could not find any clarification and data or evidence related to this classification in the manuscript. For oral conditions to be the exposure, ideally, we need evidence from prospective cohort studies in which the incidence of NCDs was compared between individuals with the oral condition and those without the oral condition after a long-term following up of individuals without the NCD at baseline. Similarly, for NCD to be the exposure, we need to compare the outcome (incidence of oral diseases) between individuals with and without the NCD at the baseline.

3. If evidence from prospective, long-term cohort studies were not available or very limited, data from case-control studies or cross-sectional studies may be used, with methodological limitations. Evidence from cross-sectional studies and case-control studies are likely to be biased because of their design weaknesses and numerous known/unknown confounding factors. Particularly, associations between oral conditions and NCDs may be due to common risk factors shared by them, such as low income/education, lack of health care support, and unhealthy behaviours. In the manuscript, the authors have completely ignored the likely influences of known/unknown confounding factors on the

reported associations. It is unclear about design types of the primary studies included in the meta-analyses, and whether and how possible confounding factors were adjusted in the meta-analyses.

4. The use of fail-safe N for the possibility of future changes in evidence relies on statistical test p values, which will be invalid if the observed association was a biased estimate.

5. Effect sizes used in the manuscript need to be more clearly defined. The included meta-analyses may use effect measures (such as RR, OR, HR, RD, etc) to quantify the effect size, but were not reflected in this overview of SRs.

6. It is possible that some primary studies (and patients) may be included in different meta-analyses. Data independence in meta-analyses for an association needs to be confirmed. In addition, is it possible that some primary studies were included in both meta-analyses using the oral condition as an outcome and meta-analyses using it as an exposure?

7. Publication years (and literature search periods) of the included meta-analyses should be explicitly reported, to reflect how updated were the available evidence.

8. Discussion page 12 line 209: "Fifty-nine associations were considered of strong evidence, supported by highly significant results and an absence of bias". How can the authors be so certain about "an absence of bias"? A good quality systematic review may be used to reveal but in general unable to confidently correct biases in primary studies (such as most case-control and cross sectional studies).

9. Throughout the manuscript (ie, page 12), the authors emphasized the nature of bidirectional association between oral conditions and NCDs, which may be biologically plausible. However, evidence summarised in this manuscript were on associations (and likely biased associations). Association and causation should be explicitly distinguished.

We are grateful to the reviewers for their insightful comments on the revised manuscript and with the opportunity to revise and resubmit once again our paper (ID number NCOMMS-22-08542B).

We have been able to incorporate changes to reflect all the suggestions provided by the reviewers. We have highlighted the changes within the manuscript. Here is a point-by-point response to the reviewers' comments and concerns.

REVIEWER COMMENTS

Reviewer #1 (Remarks to the Author):

The authors have now addressed the reviewers comments and the paper is much improved as a consequence. The quality of the English is however still very mixed and needs a comprehensive edit.

Our answer: We appreciate the reviewer's assessment. Accordingly, we have revised the English through a comprehensive edit.

Reviewer #2 (Remarks to the Author):

Thank you for the carefully done revision of the manuscript. I have only minor further suggestions:

-This statement requires a reference: "Concerning tooth loss and edentulism, both are mainly clinical endpoints of mainly periodontal diseases, and to a minor extent of dental caries." The reasons for tooth loss are associated with age.

Our answer: As suggested by the reviewer, we have added a reference to the outlined sentence. We added studies from Carvalho et al. and Gerritsen et al, references number 14 and 15, respectively.

-Please, correct this and other species names: Porphyromonas Gingivalis -> Porphyromonas gingivalis

Our answer: Following your remark, we have corrected species names accordingly. We appreciate the thoughtfulness.

Reviewer #4 (Remarks to the Author):

1. This overview of systematic reviews (SRs) included 294 systematic reviews, with 856 meta-analytic comparisons, regarding associations between oral conditions and systemic diseases (NCDs). The study found strong or suggestive associations between many NCDs (including some cancers) and oral health conditions.

Our answer: Nothing to add.

2. I feel that authors' response to comments from Reviewer #3 are generally satisfactory, but not sure about the response concerning Fig 2. If my understanding is correct, the authors considered

the included meta-analyses as two categories: (1) oral conditions as outcomes or (2) as exposures. However, I could not find any clarification and data or evidence related to this classification in the manuscript. For oral conditions to be the exposure, ideally, we need evidence from prospective cohort studies in which the incidence of NCDs was compared between individuals with the oral condition and those without the oral condition after a long-term following up of individuals without the NCD at baseline. Similarly, for NCD to be the exposure, we need to compare the outcome (incidence of oral diseases) between individuals with and without the NCD at the baseline.

Our answer: We acknowledge the importance of this commentary. We have established those two categories based on the PE(I)CO framework proposed and followed by the meta-analyses included. Whenever a meta-analysis would fall in each category, it was grouped accordingly. We agree that prospective cohort studies and clinical trials provide more robust results. Yet, as emphasized by Shea et al. (2017), the selection of the study designs for inclusion in a systematic review is the ultimate step, and just focusing on a particular study design may provide an incomplete summary of the results. We further point out, in the Discussion subsection 'Strengths and Limitations', on page 17, that most meta-analyses included observational studies: "The current umbrella review is heavily based on meta-analyses of observational studies and with a low percentage of longitudinal prospective studies and randomized trials" (this sentence was updated based as well on your remark number 9).

3. If evidence from prospective, long-term cohort studies were not available or very limited, data from case-control studies or cross-sectional studies may be used, with methodological limitations. Evidence from cross-sectional studies and case-control studies are likely to be biased because of their design weaknesses and numerous known/unknown confounding factors. Particularly, associations between oral conditions and NCDs may be due to common risk factors shared by them, such as low income/education, lack of health care support, and unhealthy behaviours. In the manuscript, the authors have completely ignored the likely influences of known/unknown confounding factors on the reported associations. It is unclear about design types of the primary studies included in the meta-analyses, and whether and how possible confounding factors were adjusted in the meta-analyses.

Our answer: Thank you for pointing this out. The reviewer is correct that if evidence from prospective, long-term cohort studies is not available or very limited, data from case-control studies or cross-sectional studies may be used, with methodological limitations. We have not ignored the likely influences of known/unknown confounding factors on the reported associations, yet most meta-analyses using case-control and cross-sectional designs have not considered confounding factors.

Regarding the possible impact of design types of the primary studies included in the meta-analyses (already mentioned in remarks 2 and 9), we had originally discussed the impact of these meta-analyses were mainly based on observational evidence. This sentence was improved based on your remarks and previously said in response to remark 2, and, and you can see more in the answer to question 9.

Considering the possible impact of known/unknown confounding factors, we decided to add this as a limitation in the section 'Strengths and Limitations' that reads as follows in a new paragraph:

"One additional shortcoming is the fact that most meta-analyses produced association estimates based on unmeasured confounding 48,49. In the studies included in this review, most explored the impact of confounding factors through subgroup meta-analyses or meta-regression. These methods are limited and do not minimize the potential confounding bias in meta-analysis 49, and

the assessment of confounder adjustment strategies shall be considered in the future. To our view, this can be achieved using reported effect sizes already adjusted for confounding factors. Nevertheless, the heterogeneity and variability of confounding factors reported in studies may limit this ultimate challenge, and for this reason an overall framework to improve the reporting of adjusted estimates is recommended.”

4. The use of fail-safe N for the possibility of future changes in evidence relies on statistical test p values, which will be invalid if the observed association was a biased estimate.

Our answer: We appreciate this commentary. Considering that FSN represents the number of studies required to refute significant meta-analytic means, it is purely a mathematical estimation, yet regarding the interpretation of this reviewer that these FSN results derive from biased estimates, we respectfully disagree. While most meta-analyses have been sourced from cross-sectional and case-control studies, we cannot draw a definitive conclusion that these associations are biased merely due to the primary nature of the included studies. What we can interpret is the stability of results from these associations, that in fact is what FSN provides.

5. Effect sizes used in the manuscript need to be more clearly defined. The included meta-analyses may used effect measures (such as RR, OR, HR, RD, etc) to quantify the effect size, but were not reflected in this overview of SRs.

Our answer: We agree that the effect sizes would be important to be defined and reported. Indeed, we have reported this information in the Supplementary data 5 (excel document appended in the system), where the reviewer can find the type of effect size used in each meta-analysis. We appreciate this remark and we could not be more in accordance.

6. It is possible that some primary studies (and patients) may be included in different meta-analyses. Data independence in meta-analyses for an association needs to be confirmed. In addition, is it possible that some primary studies were included in both meta-analyses using the oral condition as an outcome and meta-analyses using it as an exposure?

Our answer: Regarding the first statement, we completely agree. In fact, it is expected that primary studies are likely to be included in different meta-analyses particularly with the update on a topic through subsequent meta-analyses throughout the years. Therefore, data independence is something that has been not considered in the field, to the best of our knowledge. Alternative approaches have been introduced to this end such as Trial Sequential Analysis (Wetterslev et al. 2008, Thorlund et al. 2009), however this recommended to be carried out in meta-analyses and not in umbrella reviews using successive results of meta-analyses within the same topic. Therefore, we decided to follow a conservative approach of not conducting procedures without proper statistical validation, and we hope this reviewer might understand this.

7. Publication years (and literature search periods) of the included meta-analyses should be explicitly reported, to reflect how updated were the available evidence.

Our answer: We agree and, once again, we have reported this information in the Supplementary data 5. This information reflects, as stated by the reviewer, how updated the available evidence was.

8. Discussion page 12 line 209: "Fifty-nine association were considered of strong evidence, supported by highly significant results and an absence of bias". How can the authors be so certain about "an absence of bias"? A good quality systematic review may be used to reveal but in general unable to confidently correct biases in primary studies (such as most case-control and cross sectional studies).

Our answer: Thank you for pointing this out. After careful consideration we decided to remove the "an absence of bias" due to the reasons presented by the reviewer that we totally agree with. The revised text reads as follows on: "Fifty-nine associations were considered of strong evidence, supported by highly significant results" (page 12, line 209).

9. Throughout the manuscript (ie, page 12), the authors emphasized the nature of bidirectional association between oral conditions and NCDs, which may be biologically plausible. However, evidence summarised in this manuscript were on associations (and likely biased associations). Association and causation should be explicitly distinguished.

Our answer: We agree with this commentary. We have previously outlined the difference between association and causation. On page 17 we stated "The current umbrella review is heavily based on meta-analyses of observational studies and with a low percentage of randomized trials. Hence, the overall view of these results relies more on non-inferential evidence than definitive causal assumptions.". To improve this statement and provide clearer distinction, we updated as follows: "The current umbrella review is heavily based on meta-analyses of observational studies and with a low percentage **of longitudinal prospective studies** and randomized trials".

REFERENCES USED

Carvalho, R. et al. Predictors of tooth loss during long-term periodontal maintenance: An updated systematic review. *J. Clin. Periodontol.* (2021) doi:10.1111/jcpe.13488.

Gerritsen, A. E., Allen, P. F., Witter, D. J., Bronkhorst, E. M. & Creugers, N. H. Tooth loss and oral health-related quality of life: a systematic review and meta-analysis. *Health Qual. Life Outcomes* 8, 126 (2010).

Shea BJ, Reeves BC, Wells G, Thuku M, Hamel C, Moran J, Moher D, Tugwell P, Welch V, Kristjansson E, Henry DA. AMSTAR 2: a critical appraisal tool for systematic reviews that include randomised or non-randomised studies of healthcare interventions, or both. *BMJ.* 2017 Sep 21;358:j4008. doi: 10.1136/bmj.j4008. PMID: 28935701; PMCID: PMC5833365.

Thorlund K, Devereaux PJ, Guyatt G, et al. Can trial sequential monitoring boundaries reduce spurious inferences from meta-analyses?, *Int J Epidemiol*, 2009, vol. 38 (pg. 276-86)

Wetterslev J, Thorlund K, Brok J, Gluud C. Trial sequential analysis may establish when firm evidence is reached in cumulative meta-analysis, *J Clin Epidemiol*, 2008, vol. 61 (pg. 64-75)

REVIEWER COMMENTS

Reviewer #4 (Remarks to the Author):

Thanks for authors' response to my comments. I appreciate the authors' efforts, but I am afraid I have to say that their response is not satisfactory in terms of a couple of key issues I raised.

As the overview mainly included meta-analyses of observational studies, the consideration of known and unknown confounding factors is essential to obtain valid results. The authors mentioned (to comment point 3) that "yet most meta-analyses using case-control and cross-sectional designs have not considered confounding factors". This means that confounding factors might have been considered in some meta-analyses. Then we would like to know whether results of meta-analyses that considered confound factors were different from those without considering confounding factors (for the same association estimated).

The authors mentioned that "the studies included in this review, most explored the impact of confounding factors through subgroup meta-analyses or meta-regression." In primary observational studies, confounding factors can be adjusted using multi-variate regression analyses individual participant data subgroup analyse. Adjustment of confounding factors in primary studies should have been considered in well conducted and reported, or true high quality, meta-analyses of observational studies. If most published meta-analyses of observational studies did not consider confounding factors, it indicates poor quality of such meta-analyses. Without explicitly assessing and reporting confounding problems in the included meta-analyses (and primary studies they included), the authors of this overview missed an opportunity of pointing out what should be improved in systematic reviews.

In response to my comment point 4, the authors stated that "we cannot draw a definitive conclusion that these associations are biased merely due to the primary nature of the included studies." My question is how confident the authors can draw a definitive conclusion that these associations are NOT biased"? Because of well known publication/reporting bias (i.e., selective reporting of positive or significant results for publication), it is more likely that results in observational studies are biased. Please see, for example, "Why most published research findings are false" <https://pubmed.ncbi.nlm.nih.gov/16060722/> . Burden of proof should be basically on authors of primary studies, but systematic reviewers need to assess whether results of primary studies included in meta-analyses were biased. Without proper assessment of risk of bias of primary studies, we cannot assume results of meta-analyses or umbrella meta-analyses were valid.

The authors' response to my comment point 6 is not satisfactory. Data independence is a basic condition for valid statistical analysis. In this context, statistical pooling of studies in meta-analyses should avoid the multiple use of identical data from the same studies for estimating an association. The authors acknowledged that "it is expected that primary studies are likely to be included in different meta-analyses particularly with the update on a topic through subsequent meta-analyses throughout the years. Therefore, data independence is something that has been not considered in the field." Given this

lack of data independence, the pooled P values reported in this manuscript were no longer statistically valid! Therefore, the main conclusion of the manuscript, based on fail-safe N statistics, is misleading and incorrect.

Note that the problem of lack of independence cannot be resolved by using trial sequential analyses (mentioned in response to comment point 6).

The authors provided data on effect measures (response to comment point 5) and publication years (response to comment point 7) in supplementary files. The information is essential and should be properly summarised and described in the main text.

The authors' response to other points are acceptable.

Thank you for giving us the opportunity to submit the revised draft of the manuscript for publication in Nature Communications (ID number NCOMMS-22-08542D).

We once again appreciate the time and effort dedicated to providing feedback on our manuscript and are grateful for the insightful comments.

We assessed and addressed all commentaries provided by referee 4. Those changes are highlighted within the manuscript. Please see below, in blue, for a point-by-point response to the reviewers' comments and concerns. All page numbers refer to the revised manuscript file with tracked changes.

REVIEWER COMMENTS

Reviewer #4 (Remarks to the Author):

Thanks for authors' response to my comments. I appreciate the authors' efforts, but I am afraid I have to say that their response is not satisfactory in terms of a couple of key issues I raised.

As the overview mainly included meta-analyses of observational studies, the consideration of known and unknown confounding factors is essential to obtain valid results. The authors mentioned (to comment point 3) that "yet most meta-analyses using case-control and cross-sectional designs have not considered confounding factors". This means that confounding factors might have been considered in some meta-analyses. Then we would like to know whether results of meta-analyses that considered confound factors were different from those without considering confounding factors (for the same association estimated).

The authors mentioned that "the studies included in this review, most explored the impact of confounding factors through subgroup meta-analyses or meta-regression." In primary observational studies, confounding factors can be adjusted using multivariate regression analyses individual participant data subgroup analyse. Adjustment of confounding factors in primary studies should have been considered in well conducted and reported, or true high quality, meta-analyses of observational studies. If most published meta-analyses of observational studies did not consider confounding factors, it indicates poor quality of such meta-analyses. Without explicitly assessing and reporting confounding problems in the included meta-analyses (and primary studies they included), the authors of this overview missed an opportunity of pointing out what should be improved in systematic reviews.

Our answer: Out of all included systematic reviews with meta-analysis, 24.2% of the included systematic reviews did analyses with confounding variables, mostly with subgroup analyses that were developed to explore heterogeneity sources. However, the results of meta-analyses that considered confounding factors did not differ from those without considering confounding factors, particularly in those with strong meta-analytical evidence. Nevertheless, we followed the valid suggestion of "pointing out what should be improved in systematic reviews", and so we added this in a new phrase that reads as follows:

"Thus, future systematic reviews, and to a greater extent primary studies, shall improve data reporting by providing effect sizes adjusted for significant confounding factors and therefore increasing the consistency of results and preventing residual confounding" (Page 18, Paragraph 2).

In response to my comment point 4, the authors stated that "we cannot draw a definitive conclusion that these associations are biased merely due to the primary nature of the included studies." My question is how confident the authors can draw a definitive conclusion that these associations are NOT biased"? Because of well known publication/reporting bias (i.e., selective

reporting of positive or significant results for publication), it is more likely that results in observational studies are biased. Please see, for example, “Why most published research findings are false” <https://pubmed.ncbi.nlm.nih.gov/16060722/>. Burden of proof should be basically on authors of primary studies, but systematic reviewers need to assess whether results of primary studies included in meta-analyses were biased. Without proper assessment of risk of bias of primary studies, we cannot assume results of meta-analyses or umbrella meta-analyses were valid.

Our answer: We appreciate this commentary.

1. When this reviewer referred to “umbrella meta-analyses”, to be clear, we have not conducted meta-analyses in this overview, that is we assessed the reported evidence of each systematic review and did not analyze raw data.
2. With respect to the risk of bias of primary studies, the validity of systematic reviews (and its respective meta-analysis) are indeed dependent on the application of a valid risk of bias tool. This was assessed and reported in this study using the AMSTAR2 that accounts for this on item 9 (Did the review authors use a satisfactory technique for assessing the risk of bias (RoB) in individual studies that were included in the review?). In addition, it also accounted for the impact of rob on the meta-analysis per se on item 12 (If meta-analysis was performed, did the review authors assess the potential impact of RoB in individual studies on the results of the meta-analysis or other evidence synthesis?) and on the discussion and interpretation of the results on item 13 (Did the review authors account for RoB in individual studies when interpreting/ discussing the results of the review?). For this reason, the Methodological quality assessment was a mandatory step of this umbrella review and duly reported and discussed.
3. Also, on the example provided, that we appreciate, the interpretation of this study requires context within the era when it was published. At that time, reporting guidelines were scarce and highly required, with this absence being highly detrimental and with implications for the conduct and interpretation of research. Nevertheless, the subsequent years came with several important reporting guidelines developed and published with the collaboration of the same author of this example. Guidelines such as PROSPERO, TRIPOD, STARD, STROBE and PRISMA (used in this overview and employed in 77.8% of the systematic reviews included in the present study). These guidelines were developed to increase consistency and to positively impact future methodological quality analyses in systematic reviews.

The authors’ response to my comment point 6 is not satisfactory. Data independence is a basic condition for valid statistical analysis. In this context, statistical pooling of studies in meta-analyses should avoid the multiple use of identical data from the same studies for estimating an association. The authors acknowledged that “it is expected that primary studies are likely to be included in different meta-analyses particularly with the update on a topic through subsequent meta-analyses throughout the years. Therefore, data independence is something that has been not considered in the field.” Given this lack of data independence, the pooled P values reported in this manuscript were no longer statistically valid! Therefore, the main conclusion of the manuscript, based on fail-safe N statistics, is misleading and incorrect.

Note that the problem of lack of independence cannot be resolved by using trial sequential analyses (mentioned in response to comment point 6).

Our answer: After careful reading your commentary we believe our initial interpretation and answer may have misled this reviewer. This umbrella review did not use raw data from the included studies, but rather analyzed the reported evidence from each one. We had analyzed whether multiple use of identical data from the same studies in meta-analyses occurred. Our extensive analysis concluded that 3.74% (n=11) meta-analyses had possible reasons for

interpreting as data dependence, yet the final results from the present study regarding strong evidence did not change. Most studies in this situation both present critically low methodological quality and weak meta-analytical evidence, highly expected under the circumstances.

In order to avoid further misinterpretations, we have rephrased the statement:

“It is important to note that data independence is an elementary condition for valid statistical analysis. For instance, when data from the same patients’ cohort is published in different papers, care should be taken to avoid data duplicity during data pooling. Ideally, samples from the same cohort published in different articles should be grouped under one study name and have their data reported together, based on the original characteristics of the sample. This will preclude the multiple use of identical data from the same studies for estimating an association. Therefore, the interpretation of meta-analysis where “identical / overlapping data” could be identified should be interpreted with caution, as the pooled P values reported in the original review are not valid (this may also affect the interpretation and conclusions of a review). In this umbrella review, data dependence was found in 3.74% of the meta-analyses, a residual value matched by low-graded methodological quality and weak evidence strength.” (Page 18, Paragraph 3).

The authors provided data on effect measures (response to comment point 5) and publication years (response to comment point 7) in supplementary files. The information is essential and should be properly summarised and described in the main text.

Our answer: We thank this reviewer for recognizing that such data has been provided in supplementary files. We have summarized and described both information in the main text as suggested. On page 5 we added new pieces of information that read as follows:

“Mean Difference (31.9%, n=273), Odds Ratio (28.4%, n=243), Risk Ratio (16.0%, n=137) and Standardized Mean Difference (14.5%, n=124) were the most common reported effect measures (Supplementary Data 2). Out of the 294 studies, 69.0% (n=203) were published between 2011 and 2020, while 26.2% (n=77) were published in 2021 and 2022, and 4.8% (n=14) until 2010. About 24.5% (n=72) had a search period limit of 2020 to 2022.”.

The authors’ response to other points are acceptable.

Our answer: We are glad that the remaining responses were acceptable.

REVIEWERS' COMMENTS

Reviewer #4 (Remarks to the Author):

Thanks for the revised manuscript. In general, I found that the authors' response and corresponding changes are acceptable. I have no more comments to offer.

REVIEWER COMMENTS

Reviewer #4 (Remarks to the Author):

Thanks for the revised manuscript. In general, I found that the authors' response and corresponding changes are acceptable. I have no more comments to offer.

Our answer: We are grateful for your comments. We are happy that our response and changes were deemed acceptable.